**Data Availability Statement:** All relevant data are within the manuscript and its Supporting Information files.

**Funding:** The authors received no specific funding for this work.

# Productivity, nutrient use efficiency, energetic, and economics of winter maize in south India

**Siddharth Hulmani[1]⊙, S. R. Salakinkop[iD][2]⊙*, G. Somangouda[3]**

**1** Department of Agronomy, College of Agriculture, Dharwad, University of Agricultural Sciences, Dharwad, Karnataka, India, **2** AICRP on Maize, Main Agricultural Research Station, University of Agricultural Sciences, Dharwad, Karnataka, India, **3** AICRP on Soybean, Main Agricultural Research Station, University of Agricultural Sciences, Dharwad, Karnataka, India

⊙ These authors contributed equally to this work.
* salakinkopsr@uasd.in

## Abstract

The winter maize area is rapidly spreading in south India in response to rising demand from the poultry and fish feed industries due to the absence of major environmental constraints. Further farmers' are using the winter environment to expand maize area and production. Hence there is immense potential to increase the area under winter maize cultivation. There were no planned field experiments to explore and optimize the right time of sowing and quantity of fertilizer to be added previously due to the presence of negligible winter maize area. Farmers used to cultivate maize as per their choice of sowing time with the application of a quantity of fertilizer recommended for rainy season maize. There were no efforts made towards working on economic analysis including energy budgeting. And hence the investigation was conducted with the objective to explore the optimal planting period and fertilizer levels for winter maize through economic and energy budgeting. Planting windows (1st week of October, 2nd week of October, 3rd week of October, 4th week of October, and 5th week of October) and fertility levels (100 percent recommended dose of fertilizer (RDF), 150 percent RDF, and 200 percent RDF) were used as factors in Factorial Randomized Complete Block Design (RCBD) with three replications. The present investigation revealed that significantly higher winter maize productivity was achieved from the first and second week of October planting along with the application of 200% RDF (recommended dose of fertilizer) followed by 150% RDF. Planting of winter maize during the first week of October recorded significantly higher grain yield (8786kg ha$^{-1}$) and stover yield (1220 kg ha$^{-1}$) and was found on par with sowing during the second week of October. Among fertility levels, significantly higher grain yield (8320 kg ha$^{-1}$) and stover yield (1195 kg ha$^{-1}$) was recorded with the application of 200% RDF and were found on par with the application of 150% RDF. Further interaction effect showed that higher dry matter production, more days for physiological maturity, higher accumulation of growing degree days, photothermal units, and heliothermal units were recorded from crops planted during the first and second week of October along with the application of either 200% or 150% RDF. However, higher nutrient use efficiency was recorded from the first and second week of October planted crop supplied with lower fertility level (100% RDF). Similarly, significantly higher net returns and gross returns, output energy, net energy, and specific energy were higher from crops planted during the first week

**Competing interests:** The authors have declared that no competing interests exist.

of planting along with the application of 200% RDF. Whereas, energy use efficiency and energy productivity were higher with the first week of October planted crop applied with 100% RDF. From the overall interaction, it is recommended to plant winter maize during the first fortnight of October with the application of 150 percent RDF for sustaining higher maize productivity, energy output, and economics in the maize growing area of south India.

## Introduction

The world's maize area is 192.50 million hectares, and it ranks first in production with 1,112.40 million metric tonnes. The leading producers are USA (32.61%), followed by China (22.91%), Brazil (9.42%), European Union (8.41%), Argentina (5.41%) and India (4.1%) [1]. After rice and wheat, maize is India's third most popular crop. It is currently grown on 9.38 million hectares with a yield of 28.752 million metric tons [2]. Because of its photo-thermo-insensitive nature and highest genetic yield potential among cereals, maize is known as the "Queen of Cereals." Maize is grown all year round in India, in most states, for a variety of purposes including food, feed, fodder, green cobs, sweet corn, baby corn, popcorn, and industrial goods. There are three distinct seasons for the cultivation of maize in India *viz.*, rainy, winter season in peninsular India and Bihar, and spring in northern India. Maize is predominantly a rainy season crop but in past few years, winter maize has gained a significant place in total maize production in India [3]. Winter maize is grown on an area of 1.697 m ha with a production of 8.302 million metric tons and with a productivity of 4893 kg ha$^{-1}$ [4]. The predominant winter maize growing states are Bihar (26.3%), Tamil Nadu (13.1%), Maharashtra (12.9%), West Bengal (12.4%), Andhra Pradesh (9.5%), Telangana (6.9%), and Karnataka (6.4%) [4]. It has emerged as an important crop in non-traditional areas. While the crop responds favorably to better crop management in both the rainy and winter seasons, the irregular rainfall pattern of the southwest monsoon interferes with timely field operations of the rainy season. Due to the lack of significant environmental impediments in winter, the desired field operations can be scheduled and carried out at the most suitable time. In addition, the lack of any major diseases and insect pests in this season is helping the crop to express its potential. There is therefore an enormous opportunity to increase the area under cultivation of winter maize for higher productivity [3]. The increased productivity shows the higher nutrient use efficiency. The better planting date will provide a congenial environment for plants to uptake more nutrients so that the productivity of crops is increased. Nutrients use efficiency (NUE) shows the ability of crops to take up and utilize nutrients for higher productivity [3, 4]. NUE depends on the plant's ability to take up nutrients efficiently from the soil but also depends on internal transport, storage, and remobilization of nutrients [5]. NUE of applied fertilizers may be very low due to many reasons like surface runoff, leaching, volatilization, denitrification, and fixation in the soil. In agriculture development, the energy audit of various resources plays a key role in resource management. The changing global climatic conditions and increasingly growing, energy demands necessitate the development of a production system that utilizes less energy and produces more energy as output. Fertilizer as input had the highest rate of energy equivalency of all the inputs used in maize production at 51.5 percent [6]. The findings of Khokhar et al. [7] recorded higher input energy, output energy, and energy balance in higher fertility levels and higher energy use efficiency and energy productivity in lower fertility levels. In both the rainy and winter season, higher gross returns and net returns were obtained from maize sown early in the season compared to late sown crops [8, 9].

For winter maize, the best sowing date is essential so that the genotype grown can complete its life cycle under ideal environmental conditions. Planting at the beginning of the growing season is generally recommended. Sowing at the right time is critical for maximum yield, as a delay in the planting date would result in a linear decrease in grain and stover yields [10]. The amount of yield reduction caused by delayed sowing, on the other hand, varies by location. Hence experiment was conducted to explore the most congenial sowing period in Southern India. The availability of sufficient nutrients in the soil in an available form for plant uptake determines crop plant growth and yield [11]. Several other factors influence winter maize production and productivity; however, fertilizer management is one of the most important factors influencing maize growth and yields [12]. Early maize planting can improve grain yields significantly, but other practices such as fertility can also trigger the yield. Fertilizer management or recommendations on time and method of application of fertilizers have already existed and it is similar to rainy season maize. However rate of application needs to be worked out to meet higher winter maize productivity.

In agriculture, energy usage has intensified as a result of increasing population, a limited supply of arable land, and a demand for a higher standard of living [13]. It is critical to create a production system that uses less energy and produces more energy as output in the context of changing global climatic conditions and increasing energy demands [14]. In this experiment, all of the inputs and outputs were considered to be in the form of energy. Human labor, animal power, fertilizer, gasoline, and electricity are all used in some way in agricultural operations [15]. All the inputs supplied and the output obtained are considered in the form of energy. Power, electricity, machinery, seeds, fertilizers, and chemicals account for a significant portion of the existing agricultural production system's energy supply. One of them is fertilizer control with care. Since, on the one hand, it accounts for more than half of the total input energy used in the maize production system in many cases, and on the other hand, it is the most important factor for proper plant growth and development. However, if it is used excessively in such cases, it can pollute rivers and streams, as well as cause greenhouse gas emissions. Fertilizers, with an energy equivalency of 51.5 percent, were found to have the highest rate of energy equivalency among all the inputs used in maize production [6]. Excessive application of fertilizer results in waste of resources and money while also exacerbating environmental problems [16]. The planting window is one of the non-monetary inputs. Planting a crop at the right time increases not only the biological yield but also the profitability. It is important to investigate the required level of fertility and planting window to achieve long-term climate-resilient sustainability in winter maize production. With the above backdrop, the testing hypothesis was initiated with the objective of the study of the effect of sowing windows and fertility levels on the growth and productivity of winter maize. And to work out the energy flow and production economics of winter maize cultivation under different sowing windows and fertility levels. So that recommendation on the choice of sowing windows and quantity of fertilizer to be applied winter season maize to sustain the productivity of winter maize.

## Material and methods

### Experimental site

During the winter season 2019–2020, a field experiment was conducted to investigate the response of winter maize to planting windows and fertility levels at the University of Agricultural Sciences, Dharwad (Karnataka), which is located at 15˚26' N latitude and 75˚07' E longitude with an altitude of 678 m above mean sea level (MSL). The research station is located in the Northern Transitional Zone (Zone-8), which is located halfway between the Western Hilly Zone (Zone-9) and the Northern Dry Zone (Zone-9) (Zone-3). The soil is classified as clay by

the USDA soil textural classification table [17]. Soil composed of course sand, fine sand, silt and clay by 6.23, 12.66, 28.17 and 52.93 respectively (S1 Table in S1 File). The pH of the soil was 7.6, which was neutral. Available nutrients such as nitrogen were low (261 kg ha$^{-1}$) and phosphorous (31.5 kg ha$^{-1}$) and potassium (289 kg ha$^{-1}$) were medium.

## Treatment details

The experiment was conducted in a Factorial Randomized Complete Block Design (RCBD) with three replications and fifteen treatment combinations. Planting windows (First to fifth dates of sowing were 05-10-2019, 12-10-2019, 18-10-2019, 28-10-2019, and 06-11-2019 respectively) and fertility levels (100 percent recommended dose of fertilizer (RDF) which was used as control, 150 percent RDF, and 200 percent RDF) were used as factors in the experiment. The single cross maize hybrid used was Monsanto's 900 M Gold.

## Cultivation method

To bring the soil to a fine tilth, the field was plowed once, followed by tillage with a cultivator, and harrowed twice. After the previous crop was harvested, weeds and remaining residues were removed from the experimental field. The plots were set out according to the experiment's layout design. Seeds were planted at a 60 cm x 20 cm spacing with a seed rate of 20 kg ha-1. With the aid of a marker, the lines were opened, and the seeds were hand-dibbled at a depth of 4–5 cm before being covered with soil. All other treatment plots, including control plots, had well-decomposed farmyard manure (FYM) at 10 t ha$^{-1}$ incorporated into soil two weeks before planting. The nitrogen, phosphorus, and potassium were applied at 150 kg N ha$^{-1}$, 65 kg $P_2O_5$ ha$^{-1}$, and 65 kg $K_2O$ ha$^{-1}$ for a fertility level of 100% RDF. Similarly, for 150% RDF, 225 kg N, 97.5 kg $P_2O_5$, and 97.5 kg $K_2O$ ha$^{-1}$ were applied. And for 200% RDF, 300 kg N, 130 kg $P_2O_5$, and 130 kg $K_2O$ ha$^{-1}$ were applied. Urea, diammonium phosphate (DAP), and muriate of potash (MOP) were used as a source of nitrogen, phosphorus, and potassium respectively. Zinc and iron were applied in the form of $FeSO_4$ and $ZnSO_4$ at 25 kg ha$^{-1}$. First, inter cultivation was done 30 days after planting. Hand weeding was done 30 and 55 days after planting to check the weed growth and to keep the plots free from weeds during the cropping period on all the dates of planting. With a spray of emamectin benzoate at 0.3 g liter$^{-1}$ of water twice at 20 and 40 days after sowingthe crop was protected against fall armyworm and stem borer. A spray solution of 500l ha$^{-1}$ was used at each time. For all planting dates, the rainfall obtained during respective planting windows provided ample soil moisture for germination, emergence, and early establishment of seedlings. Rainfall fell during the crop growth cycle in October (323.2 mm) and November (21.0 mm), and the rest of the season's crop was irrigated using the critical stage method. Since the experiment was conducted entirely under irrigated conditions, the crop did not experience moisture stress during the growing season.

## Growth parameters related to weather

Various thermal indices including growing degree days (GDD), photothermal index (PTI), and heliothermal units (HTU) for maize were calculated by using standard methods to know their influence on maize productivity. GDD, PTU, and HTU units can be used to assess the suitability of a region for the production of a particular crop, estimate the growth stages of crops, predict maturity, best timing of fertilizer or pesticide application, estimate the heat stress on crops and plan of planting dates [18–20]. In present study they were calculated to know their influence on maize productivity.

**Growing degree day (GDD).** Growing days were determined in this study by simply adding up daily mean air temperatures above a given threshold or base temperature [18]. It can be

expressed mathematically as follows:

$$GDD\ (°C) = \frac{Tmax + Tmin}{2} - Tb$$

Where,            $T_{max}$ - Maximum temperature (°C)
                  $T_{min}$ - Minimum température (°C)
                  $T_b$ - Base temperature 10°C [18].

**Photothermal units (PTU).** The photothermal units for a specific day represent the product of GDD and the length of the day. Photothermal units were calculated by using the equation given by Wilsie [19].

PTU (°C day hr) = GDD X L
Where, GDD–Growing degree days (°Cday)
L–Day length (hrs)

**Heliothermal Units (HTU).** The heliothermal units for a specific day are the product of multiplying GDD by the number of hours of bright sun that day. The tape in the Campbell-Stroke sunshine recorder burns when the strength of sunlight reaches a pre-determined threshold. The burn trace's total duration is equal to the number of bright sunlight hours [20]. The formula was used to measure the total HTU for each phenophase's length

Accumulated HTU (°C day hr) = GDD × Bright sunshine hours (hrs)

**Nutrient use efficiency.** The sum of products produced per unit of resource used is referred to as NUE. The amount of dry matter generated per unit of nutrient applied or absorbed is the mean nutrient efficiency. NUE is the difference between a genotype's yield on deficient soil and its yield at optimum nutrition [21]. Agronomic efficiency, physiological efficiency, and recovery efficiency [22] are the three types of nutrient efficiency that were worked out to know the relation between maize yields in response to nutrients applied.

Agronomic efficiency (AE) is defined as the economic production obtained per unit of nutrient applied. It was be calculated with the help of the following equation.

$$AE\ (kg\ kg^{-1}) = \frac{\text{Grain yield in fertilized crop (kg)} - \text{Grain yield in unfertilized crop(kg)}}{\text{Quantity of nutrient applied (kg)}}$$

Physiological efficiency (PE) indicates grain yield increase in kg per kg nutrient uptake [23]. And expressed in kilogram per kg (kg kg$^{-1}$).

$$PE\ (kg\ kg^{-1}) = \frac{\text{Grain yield of F plot} - \text{Grain yield of A plot}}{\text{Nutrient uptake of the F plot} - \text{Nutrient uptake of the A-plot}}$$

Where F–Fertilized plot; A–Unfertilized control plot

Recovery efficiency (RE) is the quantity of nutrients taken up by the crop to the per unit of nutrient applied [23] and expressed as a percentage.

$$RE\ (\%) = \frac{\text{Nutrient uptake of the F plot} - \text{Nutrient uptake of the A-plot}}{\text{Quantity of nutrients applied}} \times 100$$

**Economics.** The price in USD of the inputs prevailed at the time of their use was considered for working out the cost of cultivation per hectare treatment wise and expressed in USD ha$^{-1}$. Land preparation, inter cultivation, all applied fertilizers, FYM, seed, plant protection chemicals, irrigation, men and women wages right from sowing to harvesting, drying,

processing, and marketing of products were included for working out the cost of inputs. A gross return per hectare was calculated by taking into consideration the price of the product that prevailed in the market after harvest and grain yield per hectare and expressed in USD per hectare (USD ha$^{-1}$). The net return per hectare was calculated treatment-wise by subtracting the total cost of cultivation from gross return and expressed in USD per hectare (USD ha$^{-1}$).

Net return (USD ha$^{-1}$) = Gross return (USD ha$^{-1}$)–Cost of cultivation (USD ha$^{-1}$)

The benefit-cost ratio was calculated as follows.

$$\text{Benefit-cost ratio (B-C)} = \frac{\text{Gross return (USD ha}^{-1})}{\text{Cost of cultivation (USD ha}^{-1})}$$

Note- Indian rupee was converted to USD at 70 INR for 1 USD.

**Energetics.** All the agricultural inputs such as seeds, fertilizers, labor, animals, electricity, machinery, organic manures, *etc*., and all the agricultural outputs such as grain and straw have their equivalent energy (Mega Joules) values (Table 1). The energy balance in terms of net energy, energy use efficiency, energy productivity, and specific energy was calculated using the data on input energy, output energy using the following formulae [24–40].

Net energy (MJ ha$^{-1}$) = Total output energy (MJ ha$^{-1}$)–Total input energy (MJ ha$^{-1}$)

$$\text{Energy use efficiency} = \frac{\text{Total output energy (MJ ha}^{-1})}{\text{Total input energy (MJ ha}^{-1})}$$

$$\text{Energy productivity (kg MJ}^{-1}) = \frac{\text{Maize grain yield (kg ha}^{-1})}{\text{Total input energy (MJ ha}^{-1})}$$

**Table 1. Energy equivalents (MJ unit$^{-1}$) were used for energy input and output calculations.**

| Particulars | Unit | Energy equivalent | References |
|---|---|---|---|
| **I. Inputs** | | | |
| 1. Human labor | | | |
| Adult man | hr | 1.96 | Barut et al. [29], Kumar et al. [30], Shahin et al. [31], Yadav et al. [32] |
| Adult women | hr | 1.57 | Devi et al. [33] |
| 2. Bullocks | Pair-hr | 10.1 | Binning et al. [26], Gopalan et al. [25], Mittal et al. [24], Singh [27] and Subbian et al. [28] |
| 3. Fuel and machinery | | | |
| Diesel | 1 liter | 56.31 | Barut et al. [29], Kumar et al. [30], Shahin et al. [31], Singh et al. [34], Yadav et al. [32] |
| Tractor | hr | 62.7 | Singh et al. [34] |
| 4. Manures and fertilizers | | | |
| Farm Yard Manure | 1 t | 303.1 | Avval Mousavi et al. [35] |
| Nitrogen | kg | 60.6 | Singh et al. [34] |
| Phosphorus | kg | 11.1 | Singh et al. [34] |
| Potassium | kg | 6.7 | Singh et al. [34] |
| Zinc sulfate | kg | 20.9 | Nassiri and Singh [36] |
| 5. Maize seeds | kg | 15.2 | Rahman and Rahman [37], Yadav et al. [32] |
| 6. Insecticide | kg | 120 | Kumar et al. [30], Shahin et al. [31], Shahan et al. [38] |
| 7. Irrigation water | M$^3$ | 1.02 | Devasenapathy et al. [39] |
| **II. Outputs** | | | |
| Maize grains | kg | 14.7 | Barut et al. [29], Kumar et al. [30], Rahman and Rahman [37], Shahin et al. [31], Yadav et al. [32], Zahedi et al. [40] |
| Maize stover | kg | 12.5 | Barut et al. [29], Kumar et al. [30], Rahman and Rahman [37], Shahin et al. [31], Yadav et al. [32] and Zahedi et al. [40] |

$$\text{Specific energy (MJ kg}^{-1}) = \frac{\text{Total input energy (MJ ha}^{-1})}{\text{Maize grain yield (kg ha}^{-1})}$$

**Statistical analysis and the interpretation of data.** Fisher's method of analysis of variance, as outlined by Gomez and Gomez was used to statistically analyze the data collected at various stages of crop development [41]. The data were analyzed with the MSTAT-C statistical program, and the means were compared using the Duncan Multiple Range Test (DMRT) at a 5% level of significance. The highest mean values of all the crop and nutrient parameters statistically analyzed were denoted by the letter 'a,' which was followed by the next alphabets for lower values (b, c, d, etc.). At the 0.05 level of significance, mean values denoted by the same small letter in the column do not vary significantly.

## Results

### Descriptive statistics

During the winter cropping season (2019–2020), total rainfall of 352.0 mm was received out of which 323.2 mm was received during the planting month (October) (Fig 1 and S2 Table in S1 File). The highest and the lowest maximum temperature were 31.8°C (February) and 28.5°C (December), respectively, while the respective highest and lowest minimum temperatures were 15.5°C (January) and 20.3°C (October) respectively were recorded during the cropping period. The mean relative humidity ranged from 49.4 percent in February to 79.8 percent.

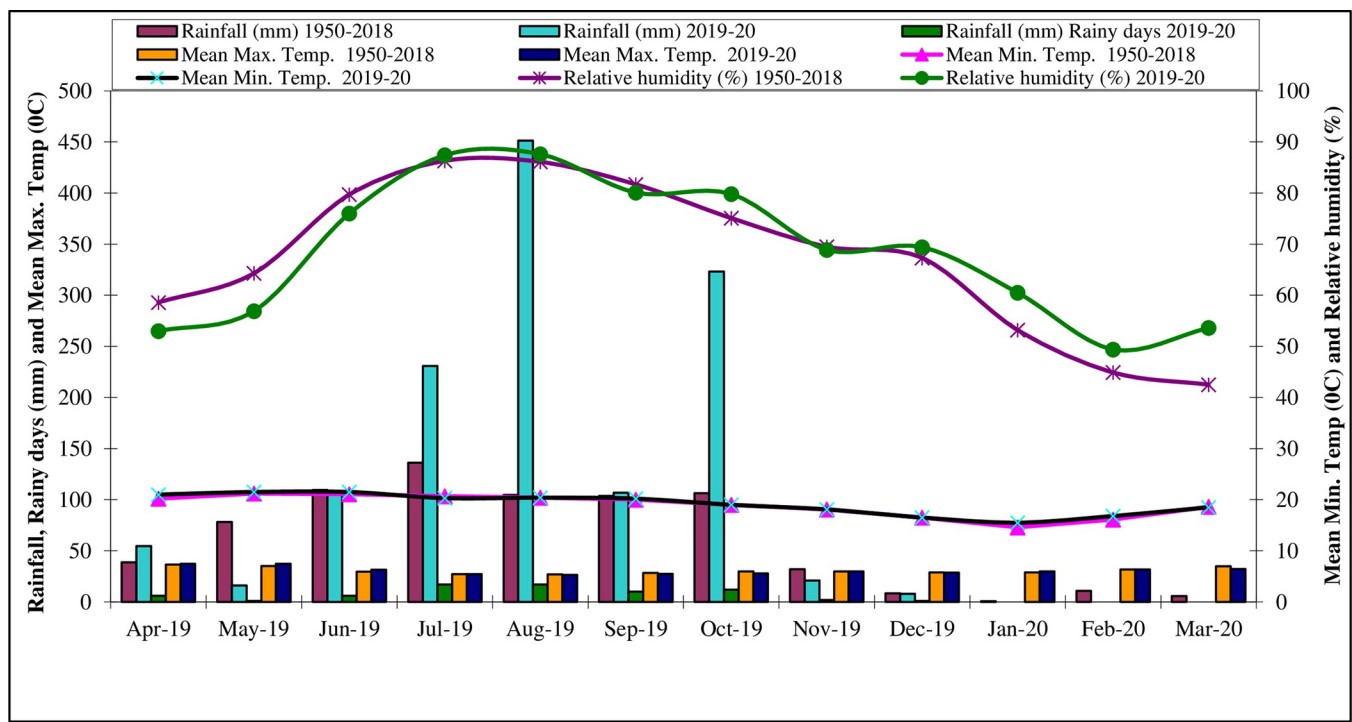

**Fig 1. Monthly meteorological data during crop growth period (2019–20) and the average of 69 years (1950–2018) at Main Agricultural Research Station, Dharwad.**

**Table 2. Days to physiological maturity, growing degree days, photothermal units, and heliothermal units of winter maize as influenced by sowing windows and fertility levels.**

| Treatment | Days to physiological maturity | Growing degree days (˚C day) | Photothermal units (˚C day hr) | Heliothermal units (˚C day hr) |
|---|---|---|---|---|
| **The factor I: Sowing windows** | | | | |
| $W_1$: 1st week of October | 120.0[a] | 1530.1[a] | 17371.3[a] | 11758.5[a] |
| $W_2$: 2nd week of October | 116.1[ab] | 1493.4[a] | 16578.0[b] | 11655.0[b] |
| $W_3$: 3rd week of October | 114.2[ab] | 1466.9[b] | 16268.3[c] | 11597.4[c] |
| $W_4$: 4th week of October | 112.0[bc] | 1444.9[c] | 15966.6[d] | 11381.6[d] |
| $W_5$: 1st week of November | 108.9[c] | 1242.4[d] | 15416.5[e] | 11274.0[e] |
| **S.Em±** | **1.26** | **2.53** | **19.33** | **12.83** |
| **Factor II: Fertility levels** | | | | |
| $F_1$: 100% RDF | 110.5[b] | 1392.7[c] | 15602.8[c] | 11149.0[c] |
| $F_2$: 150% RDF | 114.4[a] | 1421.0[b] | 16330.9[b] | 11373.3[b] |
| $F_3$: 200% RDF | 118.0[a] | 1493.5[a] | 17026.7[a] | 12077.5[a] |
| **S.Em±** | **1.63** | **3.27** | **24.95** | **16.56** |
| **Interaction (W×F)** | | | | |
| $W_1F_1$: 1st week of Oct + 100% RDF | 116.7[a-d] | 1471.4[e] | 16838.2[d] | 11409.2[f] |
| $W_1F_2$: 1st week of Oct + 150% RDF | 120.5[ab] | 1527.8[c] | 17387.3[b] | 11558.9[e] |
| $W_1F_3$: 1st week of Oct + 200% RDF | 122.9[a] | 1591.1[a] | 17888.3[a] | 12307.4[a] |
| $W_2F_1$: 2nd week of Oct + 100% RDF | 111.8[b-e] | 1444.4[f] | 15849.2[g] | 11156.4[g] |
| $W_2F_2$: 2nd week of Oct + 150% RDF | 116.3[a-d] | 1474.4[e] | 16550.3[e] | 11652.6[d] |
| $W_2F_3$: 2nd week of Oct + 200% RDF | 120.2[ab] | 1561.3[b] | 17334.6[b] | 12155.9[b] |
| $W_3F_1$: 3rd week of Oct + 100% RDF | 110.4[c-e] | 1433.4[f] | 15594.6[h] | 11351.9[f] |
| $W_3F_2$: 3rd week of Oct + 150% RDF | 114.2[a-e] | 1445.0[f] | 16193.0[f] | 11365.4[f] |
| $W_3F_3$: 3rd week of Oct + 200% RDF | 118.2[a-c] | 1522.4[c] | 17017.2[c] | 12074.7[b] |
| $W_4F_1$: 4th week of Oct + 100% RDF | 108.2[de] | 1397.8[g] | 15175.1[i] | 11012.7[h] |
| $W_4F_2$: 4th week of Oct + 150% RDF | 112.0[b-e] | 1440.0[f] | 15928.3[g] | 11170.6[g] |
| $W_4F_3$: 4th week of Oct + 200% RDF | 116.5[a-d] | 1497.0[d] | 16796.5[d] | 11961.3[c] |
| $W_5F_1$: 1st week of Nov + 100% RDF | 105.3[e] | 1213.9[i] | 14556.9[j] | 10814.6[i] |
| $W_5F_2$: 1st week of Nov + 150% RDF | 109.2[c-e] | 1218.0[i] | 15595.6[h] | 11119.0[g] |
| $W_5F_3$: 1st week of Nov + 200% RDF | 112.3[b-e] | 1295.5[h] | 16097.0[f] | 11888.3[c] |
| **S.Em±** | **2.82** | **6.67** | **43.21** | **28.6** |

The accumulated GDD, PTU, and HTU were significantly differed due to planting windows (Table 2). Planting during the 1st week of October recorded significantly higher GDD, PTU, and HTU accumulation (1530.1˚C day, 17371.3˚C day hr, and 11758.5˚C day hr respectively) and further, it took significantly more number of days for physiological maturity (120.0

days). However, it was on par with planting during the 2nd week of October (116.1 days). Similarly among the fertility levels, 200% RDF recorded significantly higher accumulated GDD, PTU, and HTU (1493.5˚C day, 17026.7˚C day hr, and 12077.5˚C day hr respectively) and application of 200% RDF took significantly more number of days for physiological maturity (118.0 days) and was on par with 150% RDF (1144.4 days). Total GDD up to maturity was sum of GDD required for each of phonological stages (S7 Table in S1 File).

The interaction effect of planting windows and fertility levels showed a significant difference in GDD, PTU, and HTU accumulation. Planting during the 1st week of October along with the application of 200% RDF recorded higher GDD, PTU, and HTU accumulation (1591.1˚Cday, 17888.3˚Cday hr, and 12307.4˚C day hr respectively. Similarly, it recorded more number days for physiological maturity.

**Grain and biomass yield.** Planting during 1st week of October recorded significantly higher grain yield and stover yield (8788 and 1220 kg ha$^{-1}$ respectively) and it was on par with planting during the 2nd week of October (8644 and 11980kg ha$^{-1}$ respectively) (Table 3 and Fig 2). As planting was delayed beyond the 2nd week of October, there was a reduction in grain and stover yield significantly. The lowest grain and stover yield was recorded from crop sown during 1st week of November. Grain and stover yield differed significantly due to fertility levels. Application of 200% RDF recorded significantly higher grain and stover yield (8320 and 11950kg ha$^{-1}$ respectively) and it was on par with the application of 150% RDF (8022 and 11410kg ha$^{-1}$ respectively).

The interaction effect of planting windows and fertility levels showed a significant difference in grain and stover yield. Planting during the 1st week of October along with the application of 200% RDF recorded significantly higher grain and stover yield (9142 and 13050kg ha$^{-1}$ respectively) and it was found on par with planting during the 2nd week of October along with the application of 200% and 150% RDF, planting during 1st week of October along with the application of 150% RDF and 100% RDF. There is no significant difference in harvest index due to planting windows and fertility levels.

## Nutrient use efficiency (NUE) in winter maize

Agronomic efficiency of nitrogen ($AE_N$), phosphorus ($AE_P$), and potassium ($AE_K$)were significantly higher with planting during 1st week of October (22.80, 52.63, and 52.63 kg kg$^{-1}$ respectively) and were on par with planting during the 2nd week of October (21.80, 48.72 and 48.72 kg kg$^{-1}$ respectively) (Table 4). Significantly higher $AE_N$, $AE_P$, and $AE_K$ were recorded with the application of 100% RDF (23.41, 53.93, and 53.93 kg kg$^{-1}$ respectively). Interaction of planting windows and fertility levels recorded significant differences in $AE_N$, $AE_P$, and $AE_K$. Planting during the 1st week of October along with the application of 100% RDF recorded significantly higher $AE_N$, $AE_P$, and $AE_K$ (29.59, 68.29, and 68.29 kg kg$^{-1}$ respectively), and it was found on par with planting during the 2nd week of October along with the application of 100% and planting during 3rd week of October along with the application of 100% RDF.

## Physiological efficiency of nutrients

Physiological efficiency of nitrogen ($PE_N$), phosphorus ($PE_P$), and potassium ($PE_K$) were higher with planting during the 1st week of November (53.80, 209.01, and 69.58 kg ha$^{-1}$ respectively). Among fertility levels, higher $PE_N$, $PE_P$, and $PE_K$ were with the application of 100% RDF (50.67, 196.68, and 69.65 kg ha$^{-1}$ respectively). Interaction effect among planting windows and fertility levels showed significant difference with$PE_N$ and significantly higher $PE_N$ was recorded during 1st week of November planting along with the application of 100% RDF (59.50 kg ha$^{-1}$). Interaction effect found non-significant concerning $PE_P$ and $PE_K$.

**Table 3. Grain yield, stover yield, and harvest index of winter maize as influenced by sowing windows and fertility levels.**

| Treatment | Grain yield (kg ha$^{-1}$) | Stover yield (kg ha$^{-1}$) | Harvest index (%) |
|---|---|---|---|
| **The factor I: Sowing windows** | | | |
| W$_1$: 1$^{st}$ week of October | 8787$^a$ | 12200$^a$ | 41.99$^a$ |
| W$_2$: 2$^{nd}$ week of October | 8644$^{ab}$ | 11980$^{ab}$ | 41.86$^a$ |
| W$_3$: 3$^{rd}$ week of October | 8133$^b$ | 11320$^{bc}$ | 41.83$^a$ |
| W$_4$: 4$^{th}$ week of October | 7436$^c$ | 10750$^{cd}$ | 41.03$^a$ |
| W$_5$: 1$^{st}$ week of November | 6732$^d$ | 10330$^d$ | 39.54$^a$ |
| **S.Em±** | **154** | **209** | **0.62** |
| **Factor II: Fertility levels** | | | |
| F$_1$: 100% RDF | 7497$^b$ | 10590$^b$ | 41.34$^a$ |
| F$_2$: 150% RDF | 8022$^a$ | 11410$^a$ | 41.34$^a$ |
| F$_3$: 200% RDF | 8320$^a$ | 11950$^a$ | 41.06$^a$ |
| **S.Em±** | **199** | **269** | **0.80** |
| **Interaction (W×F)** | | | |
| W$_1$F$_1$: 1$^{st}$ week of Oct + 100% RDF | 8403$^{a-c}$ | 11280$^{b-g}$ | 42.86$^a$ |
| W$_1$F$_2$: 1$^{st}$ week of Oct + 150% RDF | 8814$^{ab}$ | 12270$^{a-c}$ | 41.83$^a$ |
| W$_1$F$_3$: 1$^{st}$ week of Oct + 200% RDF | 9142$^a$ | 13050$^a$ | 41.26$^a$ |
| W$_2$F$_1$: 2$^{nd}$ week of Oct + 100% RDF | 8141$^{a-d}$ | 11000$^{c-g}$ | 42.23$^a$ |
| W$_2$F$_2$: 2$^{nd}$ week of Oct + 150% RDF | 8783$^{ab}$ | 12170$^{a-d}$ | 41.93$^a$ |
| W$_2$F$_3$: 2$^{nd}$ week of Oct + 200% RDF | 9007$^{ab}$ | 12780$^{ab}$ | 41.40$^a$ |
| W$_3$F$_1$: 3$^{rd}$ week of Oct + 100% RDF | 7859$^{b-e}$ | 10690$^{d-g}$ | 42.33$^a$ |
| W$_3$F$_2$: 3$^{rd}$ week of Oct + 150% RDF | 8217$^{a-d}$ | 11480$^{b-f}$ | 41.76$^a$ |
| W$_3$F$_3$: 3$^{rd}$ week of Oct + 200% RDF | 8321$^{a-c}$ | 11800$^{a-e}$ | 41.40$^a$ |
| W$_4$F$_1$: 4$^{th}$ week of Oct + 100% RDF | 69.3$^{ef}$ | 10190$^{fg}$ | 40.53$^a$ |
| W$_4$F$_2$: 4$^{th}$ week of Oct + 150% RDF | 7444$^{c-e}$ | 10780$^{c-g}$ | 41.23$^a$ |
| W$_4$F$_3$: 4$^{th}$ week of Oct + 200% RDF | 7948$^{b-e}$ | 11290$^{b-g}$ | 41.33$^a$ |
| W$_5$F$_1$: 1$^{st}$ week of Nov + 100% RDF | 6164$^f$ | 9708$^g$ | 38.76$^a$ |
| W$_5$F$_2$: 1$^{st}$ week of Nov + 150% RDF | 6850$^{ef}$ | 10380$^{e-g}$ | 39.96$^a$ |
| W$_5$F$_3$: 1$^{st}$ week of Nov + 200% RDF | 7180$^{d-f}$ | 10840$^{c-g}$ | 39.9$^a$ |
| **S.Em±** | **345** | **466** | **1.37** |
| C$_1$: Control | 3964 | 6480 | 39.30 |
| **S.Em±** | **345** | **601** | **2.25** |

Note: Highest values were denoted with 'a' followed by the next alphabets for lower values (b, c, d, *etc*.). Value denoted by the same small letter in the column does not differ significantly at 0.05 level of significance.

## The recovery efficiency of nutrients

The recovery efficiency of nitrogen (RE$_N$), phosphorus (RE$_P$), and potassium (RE$_K$) were significantly higher with planting during the 1$^{st}$ week of October (47.92, 29.32, and 78.99% respectively) and were on par with planting during 2$^{nd}$ week of October (44.36, 27.70 and 73.04% respectively). Among fertility levels, significantly higherPE$_N$, PE$_P$, and PE$_K$ were recorded with the application of 100% RDF (49.12, 31.52, and 81.23% respectively). Treatment combinations of planting windows and fertility levels recorded significant differences in RE$_N$, RE$_P$, and RE$_K$. Planting during the 1$^{st}$ week of October along with the application of 100% RDF recorded significantly higher RE$_N$, RE$_P$, and RE$_K$ (62.19, 38.43, and 102.52% respectively), and it was found on par with planting during the 2$^{nd}$ week of October along with the application of 100% and planting during 3$^{rd}$ week of October along with the application of 100% RDF.

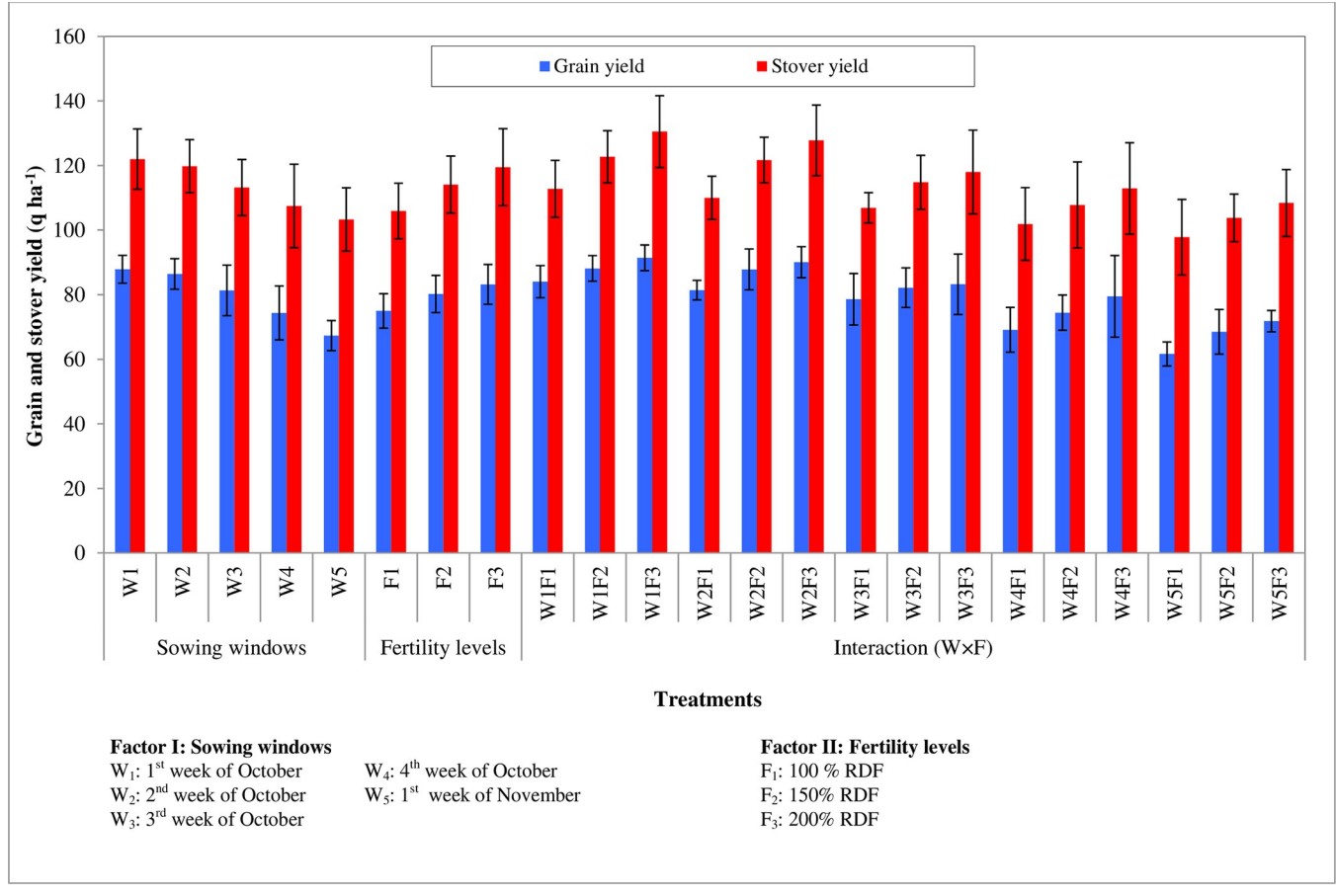

**Fig 2. Grain yield and stover yield of winter maize as influenced by sowing windows and fertility levels.**

## Energetic

Energetics of winter maize was significantly influenced by planting windows and fertility levels. The output energy, net energy, energy use efficiency, and energy productivity were significantly higher from 1st week of October planting (281233.7MJ ha$^{-1}$, 258121.4 MJ ha$^{-1}$, 12.40 and 0.39kg MJ$^{-1}$ respectively) (Table 5). However, it was found on par with planting during the 2nd week of October (276831.5 MJ ha$^{-1}$, 253719.2 MJ ha$^{-1}$, 12.19, and 0.38 kg MJ$^{-1}$ respectively). Whereas, the specific energy was significantly higher in 1st week of November planting (3.43MJ kg$^{-1}$).

Among fertility levels, significantly higher output energy, net energy, and specific energy were recorded with the application of 200% RDF (271692.2MJ ha$^{-1}$, 244035.4MJ ha$^{-1}$, and 3.36 MJ kg$^{-1}$ respectively). Whereas, the energy use efficiency and energy productivity were significantly higher with the application of 100% RDF (13.06 and 0.40 kg MJ$^{-1}$ respectively). The combination effect showed that the output energy and net energy were significantly higher in 1st week of October planting along with 200% RDF (297515.0MJ ha$^{-1}$ and 269857.8 MJ ha$^{-1}$ respectively). However, it was on par with planting during the 2nd week of October along with 200% RDF and planting during the 1st week of October along with 150% RDF. The energy use efficiency and energy productivity were significantly higher with planting during the 1st week of October along with the application of 100% RDF (14.24 and 0.45 kg MJ$^{-1}$ respectively). However, it was on par with planting during the 2nd week of October along with 100% RDF

**Table 4. Nutrient use efficiency of nitrogen, phosphorus, and potassium as influenced by sowing windows and fertility levels.**

| Treatment | $AE_N$ (kg kg⁻¹) | $AE_P$ (kg kg⁻¹) | $AE_K$ (kg kg⁻¹) | $PE_N$ (kg ha⁻¹) | $PE_P$ (kg ha⁻¹) | $PE_K$ (kg ha⁻¹) | $RE_N$ (%) | $RE_P$ (%) | $RE_K$ (%) |
|---|---|---|---|---|---|---|---|---|---|
| **The factor I: Sowing windows** | | | | | | | | | |
| $W_1$: 1st week of October | 22.80ᵃ | 52.63ᵃ | 52.63ᵃ | 49.01ᵃᵇ | 184.24ᵃ | 68.57ᵃ | 47.92ᵃ | 29.32ᵃ | 78.99ᵃ |
| $W_2$: 2nd week of October | 21.80ᵃ | 48.72ᵃᵇ | 48.72ᵃᵇ | 47.94ᵇ | 181.72ᵃ | 68.65ᵃ | 44.36ᵃ | 27.70ᵃ | 73.04ᵃᵇ |
| $W_3$: 3rd week of October | 19.80ᵃ | 45.69ᵇ | 45.69ᵇ | 48.37ᵇ | 184.45ᵃ | 66.42ᵃ | 41.59ᵃ | 26.19ᵃ | 69.09ᵇ |
| $W_4$: 4th week of October | 16.14ᵇ | 37.25ᶜ | 37.25ᶜ | 48.74ᵃᵇ | 194.31ᵃ | 69.50ᵃ | 33.92ᵇ | 21.97ᵇ | 56.01ᶜ |
| $W_5$: 1st week of November | 12.74ᶜ | 29.40ᵈ | 29.40ᵈ | 53.80ᵃ | 209.01ᵃ | 69.58ᵃ | 26.78ᶜ | 18.04ᶜ | 44.04ᵈ |
| S.Em± | 0.75 | 1.74 | 1.74 | 1.31 | 17.76 | 4.28 | 1.70 | 0.81 | 1.98 |
| Factor II: Fertility levels | | | | | | | | | |
| $F_1$: 100% RDF | 23.41ᵃ | 53.93ᵃ | 53.93ᵃ | 50.67ᵃ | 196.68ᵃ | 69.65ᵃ | 49.12ᵃ | 31.52ᵃ | 81.23ᵃ |
| $F_2$: 150% RDF | 18.03ᵇ | 41.06ᵇ | 41.06ᵇ | 49.73ᵃ | 187.26ᵃ | 68.28ᵃ | 37.39ᵇ | 23.48ᵇ | 61.65ᵇ |
| $F_3$: 200% RDF | 14.51ᶜ | 33.19ᶜ | 33.19ᶜ | 48.32ᵃ | 188.29ᵃ | 67.69ᵃ | 30.23ᶜ | 18.91ᶜ | 49.82ᶜ |
| S.Em± | 0.97 | 2.25 | 2.25 | 1.69 | 22.93 | 5.53 | 2.20 | 1.04 | 2.56 |
| $W_1F_1$: 1st week of Oct + 100% RDF | 29.59ᵃ | 68.29ᵃ | 68.29ᵃ | 47.54ᵇ | 185.96ᵃ | 66.47ᵃ | 62.19ᵃ | 38.43ᵃ | 102.52ᵃ |
| $W_1F_2$: 1st week of Oct + 150% RDF | 21.55ᵇᶜ | 49.74ᵇᶜ | 49.74ᵇᶜ | 49.54ᵇ | 185.99ᵃ | 72.13ᵃ | 45.29ᵇᶜ | 27.28ᵇ | 74.66ᵇ |
| $W_1F_3$: 1st week of Oct + 200% RDF | 17.26ᶜ⁻ᶠ | 39.83ᶜ⁻ᶠ | 39.83ᶜ⁻ᶠ | 49.95ᵇ | 180.76ᵃ | 67.10ᵃ | 36.27ᶜ⁻ᶠ | 22.22ᵇ⁻ᵉ | 59.79ᶜ⁻ᵉ |
| $W_2F_1$: 2nd week of Oct + 100% RDF | 27.17ᵃ | 62.20ᵃ | 62.20ᵃ | 47.55ᵇ | 174.68ᵃ | 70.35ᵃ | 56.64ᵃᵇ | 35.73ᵃ | 93.13ᵃ |
| $W_2F_2$: 2nd week of Oct + 150% RDF | 21.41ᵇᶜ | 46.68ᶜᵈ | 46.68ᶜᵈ | 48.66ᵇ | 184.99ᵃ | 66.97ᵃ | 42.51ᶜᵈ | 26.43ᵇᶜ | 70.08ᵇᶜ |
| $W_2F_3$: 2nd week of Oct + 200% RDF | 16.81ᶜ⁻ᶠ | 37.25ᶜ⁻ᵍ | 37.25ᶜ⁻ᵍ | 47.61ᵇ | 185.48ᵃ | 68.63ᵃ | 33.92ᶜ⁻ᵍ | 20.93ᶜ⁻ᵉ | 55.92ᶜ⁻ᶠ |
| $W_3F_1$: 3rd week of Oct + 100% RDF | 25.96ᵃᵇ | 59.92ᵃᵇ | 59.92ᵃᵇ | 49.71ᵇ | 190.37ᵃ | 65.86ᵃ | 54.57ᵃᵇ | 34.58ᵃ | 91.49ᵃ |
| $W_3F_2$: 3rd week of Oct + 150% RDF | 18.90ᶜ⁻ᵉ | 43.62ᶜ⁻ᵉ | 43.62ᶜ⁻ᵉ | 47.89ᵇ | 176.39ᵃ | 66.52ᵃ | 39.67ᶜ⁻ᵉ | 24.89ᵇ⁻ᵈ | 65.47ᵇ⁻ᵈ |
| $W_3F_3$: 3rd week of Oct + 200% RDF | 14.52ᵈ⁻ᵍ | 33.51ᵉ⁻ᵍ | 33.51ᵉ⁻ᵍ | 47.51ᵇ | 186.58ᵃ | 66.88ᵃ | 30.52ᵈ⁻ᵍ | 19.09ᵈ⁻ᶠ | 50.30ᵉ⁻ᵍ |
| $W_4F_1$: 4th week of Oct + 100% RDF | 19.67ᶜᵈ | 45.41ᶜ⁻ᵉ | 45.41ᶜ⁻ᵉ | 49.04ᵇ | 166.28ᵃ | 72.02ᵃ | 41.35ᶜᵈ | 27.33ᵇ | 68.18ᵇᶜ |
| $W_4F_2$: 4th week of Oct + 150% RDF | 15.46ᵈ⁻ᵍ | 35.69ᵈ⁻ᵍ | 35.69ᵈ⁻ᵍ | 49.33ᵇ | 215.34ᵃ | 69.45ᵃ | 32.50ᵈ⁻ᵍ | 20.92ᶜ⁻ᵉ | 53.57ᵈ⁻ᶠ |
| $W_4F_3$: 4th week of Oct + 200% RDF | 13.28ᵉ⁻ᵍ | 30.64ᶠᵍ | 30.64ᶠᵍ | 47.83ᵇ | 201.29ᵃ | 67.01ᵃ | 27.90ᵉ⁻ᵍ | 17.63ᵉᶠ | 46.27ᵉ⁻ᵍ |
| $W_5F_1$: 1st week of Nov + 100% RDF | 14.66ᵈ⁻ᵍ | 33.85ᵈ⁻ᵍ | 33.85ᵈ⁻ᵍ | 59.50ᵃ | 266.11ᵃ | 73.57ᵃ | 30.84ᵈ⁻ᵍ | 21.54ᵇ⁻ᵉ | 50.84ᵉ⁻ᵍ |
| $W_5F_2$: 1st week of Nov + 150% RDF | 12.87ᶠᵍ | 29.60ᶠᵍ | 29.60ᶠᵍ | 53.22ᵃᵇ | 173.58ᵃ | 66.32ᵃ | 26.96ᶠᵍ | 17.88ᵉᶠ | 44.45ᶠᵍ |
| $W_5F_3$: 1st week of Nov + 200% RDF | 10.72ᵍ | 24.73ᵍ | 24.73ᵍ | 48.68ᵇ | 187.33ᵃ | 68.84ᵃ | 22.53ᵍ | 14.68ᶠ | 36.82ᵍ |
| **S.Em±** | **1.68** | **3.89** | **3.89** | **2.92** | **39.71** | **9.57** | **3.80** | **1.80** | **4.43** |
| **$C_1$: Control** | - | - | - | - | - | - | - | - | - |

and planting during the 3rd week of October along with 100% RDF. The specific energy was significantly higher in 1st week of November planting along with 200% RDF (3.85MJ kg⁻¹).

## The economic value of maize grain

Significantly higher gross return, the net return, and B-C ratio were recorded with planting during 1st week of October (USD 2,331 ha⁻¹, USD 1,409 ha⁻¹, and 2.53 respectively) and it was on par with planting during 2nd week of October (USD 2,293 ha⁻¹, USD 1,371 ha⁻¹ and 2.49 respectively). Among fertility levels, significantly higher gross return, and net return were recorded with the application of 200% RDF (USD 2,211ha⁻¹ and USD 1,231 ha⁻¹ respectively). There was no significant effect observed concerning the B-C ratio among fertility levels (Table 6). Interaction effect showed that significantly higher gross return and net return were recorded with planting during 1st week of October along with application 200% RDF (USD 2,429 ha⁻¹ and USD 1,462 ha⁻¹ respectively) and it was on par with planting during 2nd week of October along with application 200% RDF (USD 2,392ha⁻¹ and USD1,425 ha⁻¹ respectively) and planting during 1st week of October along with application 150% RDF (USD 2,338 ha⁻¹

**Table 5. Energetics of winter maize as influenced by sowing windows and fertility levels.**

| Treatment | Input energy (MJ ha$^{-1}$) | Output energy (MJ ha$^{-1}$) | Net energy (MJ ha$^{-1}$) | Energy use efficiency | Energy productivity (kg MJ$^{-1}$) | Specific energy (MJ kg$^{-1}$) |
|---|---|---|---|---|---|---|
| **The factor I: Sowing windows** | | | | | | |
| W$_1$: 1$^{st}$ week of October | 23141.7 | 281233.7$^a$ | 258121.4$^a$ | 12.40$^a$ | 0.39$^a$ | 2.62$^c$ |
| W$_2$: 2$^{nd}$ week of October | 23141.7 | 276831.5$^a$ | 253719.2$^a$ | 12.19$^{ab}$ | 0.38$^{ab}$ | 2.65$^c$ |
| W$_3$: 3$^{rd}$ week of October | 23147.6 | 260631.8$^b$ | 237519.5$^b$ | 11.72$^b$ | 0.36$^b$ | 2.85$^c$ |
| W$_4$: 4$^{th}$ week of October | 23153.5 | 243285.6$^c$ | 220173.3$^c$ | 10.72$^c$ | 0.33$^c$ | 3.12$^b$ |
| W$_5$: 1$^{st}$ week of November | 23153.5 | 227703.2$^d$ | 204085.9$^d$ | 10.03$^d$ | 0.30$^d$ | 3.43$^a$ |
| **S.Em±** | - | 3552.1 | 3551.66 | 0.16 | 0.01 | 0.06 |
| **Factor II: Fertility levels** | | | | | | |
| F$_1$: 100% RDF | 18602.6 | 242543.1$^c$ | 223672.8$^b$ | 13.06$^a$ | 0.40$^a$ | 2.51$^c$ |
| F$_2$: 150% RDF | 23147.6 | 259575.7$^b$ | 236463.4$^a$ | 11.23$^b$ | 0.34$^b$ | 2.92$^b$ |
| F$_3$: 200% RDF | 27692.6 | 271692.2$^a$ | 244035.4$^a$ | 9.94$^c$ | 0.29$^c$ | 3.36$^a$ |
| **S.Em±** | - | 4585.7 | 4585.1 | 0.21 | 0.01 | 0.08 |
| **Interaction (W×F)** | | | | | | |
| W$_1$F$_1$: 1$^{st}$ week of Oct + 100% RDF | 18596.7 | 264510.1$^{c-e}$ | 245942.8$^{a-c}$ | 14.24$^a$ | 0.45$^a$ | 2.21$^h$ |
| W$_1$F$_2$: 1$^{st}$ week of Oct + 150% RDF | 23141.6 | 281676.0$^{a-c}$ | 258563.8$^{ab}$ | 12.18$^b$ | 0.38$^{bc}$ | 2.62$^{e-h}$ |
| W$_1$F$_3$: 1$^{st}$ week of Oct + 200% RDF | 27686.7 | 297515.0$^a$ | 269857.8$^a$ | 10.76$^{cd}$ | 0.32$^{ef}$ | 3.03$^{c-e}$ |
| W$_2$F$_1$: 2$^{nd}$ week of Oct + 100% RDF | 18596.7 | 257193.0$^{c-e}$ | 238625.7$^{b-d}$ | 13.85$^a$ | 0.44$^a$ | 2.28$^{gh}$ |
| W$_2$F$_2$: 2$^{nd}$ week of Oct + 150% RDF | 23141.6 | 281176.0$^{a-c}$ | 258063.7$^{ab}$ | 12.17$^b$ | 0.38$^{bc}$ | 2.64$^{e-h}$ |
| W$_2$F$_3$: 2$^{nd}$ week of Oct + 200% RDF | 27686.7 | 292125.6$^{ab}$ | 264468.4$^{ab}$ | 10.56$^{cd}$ | 0.32$^{ef}$ | 3.03$^{c-e}$ |
| W$_3$F$_1$: 3$^{rd}$ week of Oct + 100% RDF | 18602.6 | 249111.3$^{d-f}$ | 230544.0$^{c-e}$ | 13.41$^a$ | 0.42$^{ab}$ | 2.38$^{f-h}$ |
| W$_3$F$_2$: 3$^{rd}$ week of Oct + 150% RDF | 23147.6 | 262982.0$^{c-e}$ | 239869.7$^{bc}$ | 11.38$^{bc}$ | 0.35$^{c-e}$ | 2.82$^{d-f}$ |
| W$_3$F$_3$: 3$^{rd}$ week of Oct + 200% RDF | 27692.6 | 269802.0$^{b-d}$ | 242144.8$^{bc}$ | 10.36$^{c-e}$ | 0.30$^{fg}$ | 3.35$^{bc}$ |
| W$_4$F$_1$: 4$^{th}$ week of Oct + 100% RDF | 18608.5 | 228993.6$^{fg}$ | 210426.3$^{ef}$ | 12.33$^b$ | 0.37$^{cd}$ | 2.70$^{d-g}$ |
| W$_4$F$_2$: 4$^{th}$ week of Oct + 150% RDF | 23153.5 | 242888.5$^{ef}$ | 219776.2$^{c-e}$ | 10.51$^{cd}$ | 0.32$^{ef}$ | 3.11$^{b-d}$ |
| W$_4$F$_3$: 4$^{th}$ week of Oct + 200% RDF | 27698.5 | 257974.8$^{c-e}$ | 230317.5$^{c-e}$ | 9.32$^{ef}$ | 0.28$^{fg}$ | 3.53$^{ab}$ |
| W$_5$F$_1$: 1$^{st}$ week of Nov + 100% RDF | 18608.5 | 212907.3$^g$ | 192825.1$^f$ | 11.46$^{bc}$ | 0.33$^{d-f}$ | 3.02$^{c-e}$ |
| W$_5$F$_2$: 1$^{st}$ week of Nov + 150% RDF | 23153.5 | 229156.2$^{fg}$ | 206044.0$^{ef}$ | 9.91$^{de}$ | 0.29$^{fg}$ | 3.40$^{bc}$ |
| W$_5$F$_3$: 1$^{st}$ week of Nov + 200% RDF | 27698.5 | 241046.0$^{ef}$ | 213388.7$^{d-f}$ | 8.71$^f$ | 0.26$^g$ | 3.85$^a$ |
| **S.Em±** | - | 7942.7 | 7941.8 | 0.37 | 0.01 | 0.14 |
| C$_1$: Control | 5422.2 | 139228.2 | 133806.0 | 25.7 | 0.7 | 1.4 |
| **S.Em. ±** | 0.001 | 8692.8 | 8741.5 | 0.71 | 0.019 | 0.13 |

and USD 1,416 ha$^{-1}$ respectively). A significantly higher B-C ratio was recorded with planting during 1$^{st}$ week of October along with the application of 150% RDF.

**Table 6. Economics of winter maize as influenced by sowing windows and fertility levels.**

| Treatment | Cost of cultivation (USD ha$^{-1}$) | Gross return (USD ha$^{-1}$) | Net return (USD ha$^{-1}$) | B-C ratio |
|---|---|---|---|---|
| **The factor I: Sowing windows** | | | | |
| W$_1$: 1$^{st}$ week of October | 922 | 2331$^a$ | 1409$^a$ | 2.53$^a$ |
| W$_2$: 2$^{nd}$ week of October | 922 | 2293$^{ab}$ | 1371$^a$ | 2.49$^a$ |
| W$_3$: 3$^{rd}$ week of October | 935 | 2158$^b$ | 1223$^b$ | 2.31$^b$ |
| W$_4$: 4$^{th}$ week of October | 948 | 1977$^c$ | 1030$^c$ | 2.09$^c$ |
| W$_5$: 1$^{st}$ week of November | 948 | 1795$^d$ | 848$^d$ | 1.89$^d$ |
| S.Em± | - | 38.9 | 38.9 | 0.04 |
| **Factor II: Fertility levels** | | | | |
| F$_1$: 100% RDF | 889 | 1991$^b$ | 1102$^b$ | 2.24$^a$ |
| F$_2$: 150% RDF | 935 | 2130$^a$ | 1196$^{ab}$ | 2.28$^a$ |
| F$_3$: 200% RDF | 980 | 2211$^a$ | 1231$^a$ | 2.26$^a$ |
| S.Em± | - | 50.3 | 50.3 | 0.05 |
| **Interaction (W×F)** | | | | |
| W$_1$F$_1$: 1$^{st}$ week of Oct + 100% RDF | 876 | 2226$^{a-d}$ | 1350$^{ab}$ | 2.54$^a$ |
| W$_1$F$_2$: 1$^{st}$ week of Oct + 150% RDF | 922 | 2338$^{a-c}$ | 1416$^a$ | 2.55$^a$ |
| W$_1$F$_3$: 1$^{st}$ week of Oct + 200% RDF | 968 | 2429$^a$ | 1462$^a$ | 2.51$^a$ |
| W$_2$F$_1$: 2$^{nd}$ week of Oct + 100% RDF | 876 | 2157$^{a-e}$ | 1281$^{a-c}$ | 2.46$^a$ |
| W$_2$F$_2$: 2$^{nd}$ week of Oct + 150% RDF | 922 | 2330$^{a-c}$ | 1408$^a$ | 2.53$^a$ |
| W$_2$F$_3$: 2$^{nd}$ week of Oct + 200% RDF | 968 | 2392$^{ab}$ | 1425$^a$ | 2.47$^a$ |
| W$_3$F$_1$: 3$^{rd}$ week of Oct + 100% RDF | 889 | 2083$^{c-g}$ | 1194$^{a-d}$ | 2.34$^{ab}$ |
| W$_3$F$_2$: 3$^{rd}$ week of Oct + 150% RDF | 935 | 2180$^{a-e}$ | 1245$^{a-c}$ | 2.33$^{a-c}$ |
| W$_3$F$_3$: 3$^{rd}$ week of Oct + 200% RDF | 980 | 2210$^{a-d}$ | 1230$^{a-c}$ | 2.25$^{a-c}$ |
| W$_4$F$_1$: 4$^{th}$ week of Oct + 100% RDF | 902 | 1841$^{f-h}$ | 940$^{d-f}$ | 2.04$^{c-e}$ |
| W$_4$F$_2$: 4$^{th}$ week of Oct + 150% RDF | 948 | 1978$^{d-g}$ | 1031$^{c-e}$ | 2.09$^{b-e}$ |
| W$_4$F$_3$: 4$^{th}$ week of Oct + 200% RDF | 993 | 2111$^{b-f}$ | 1118$^{b-e}$ | 2.12$^{b-d}$ |
| W$_5$F$_1$: 1$^{st}$ week of Nov + 100% RDF | 902 | 1648$^h$ | 746$^f$ | 1.83$^e$ |
| W$_5$F$_2$: 1$^{st}$ week of Nov + 150% RDF | 948 | 1825$^{gh}$ | 878$^{ef}$ | 1.93$^{de}$ |
| W$_5$F$_3$: 1$^{st}$ week of Nov + 200% RDF | 993 | 1914$^{e-h}$ | 920$^{d-f}$ | 1.93$^{de}$ |
| S.Em± | - | 87.1 | 87.1 | 0.10 |
| C$_1$: Control | 746 | 1061 | 316 | 1.42 |
| S.Em± | - | 86.0 | 86.1 | 0.09 |

Note: Highest values were denoted with 'a' followed by the next alphabets for lower values (b, c, d, *etc.*). Value denoted by the same small letter in the column does not differ significantly at 0.05 level of significance. (1 USD = 70 INR).

# Discussion

## Productivity of winter maize as influenced by planting windows

The optimum date of planting is important for winter maize so that the genotype grown can complete its life cycle and express its full potential under optimum environmental conditions. For optimization of yield, planting at the appropriate time is very important as delayed planting can lead to a linear decrease in grain and stover yields [42, 52–55]. Early planting in October recorded higher productivity. Planting during October 1$^{st}$ week recorded 1.65, 8.04, 18.17, and 30.53 percent linear increase in grain yield compared to planting during 2$^{nd}$, 3$^{rd}$, 4$^{th}$ week of October and 1$^{st}$ week of November respectively (Table 3 and Fig 2). Higher grain yield obtained from October 1$^{st}$ week planting was attributed to significant improvement in yield characters and dry matter accumulation (Fig 3). Similar results were also obtained [44]. Late

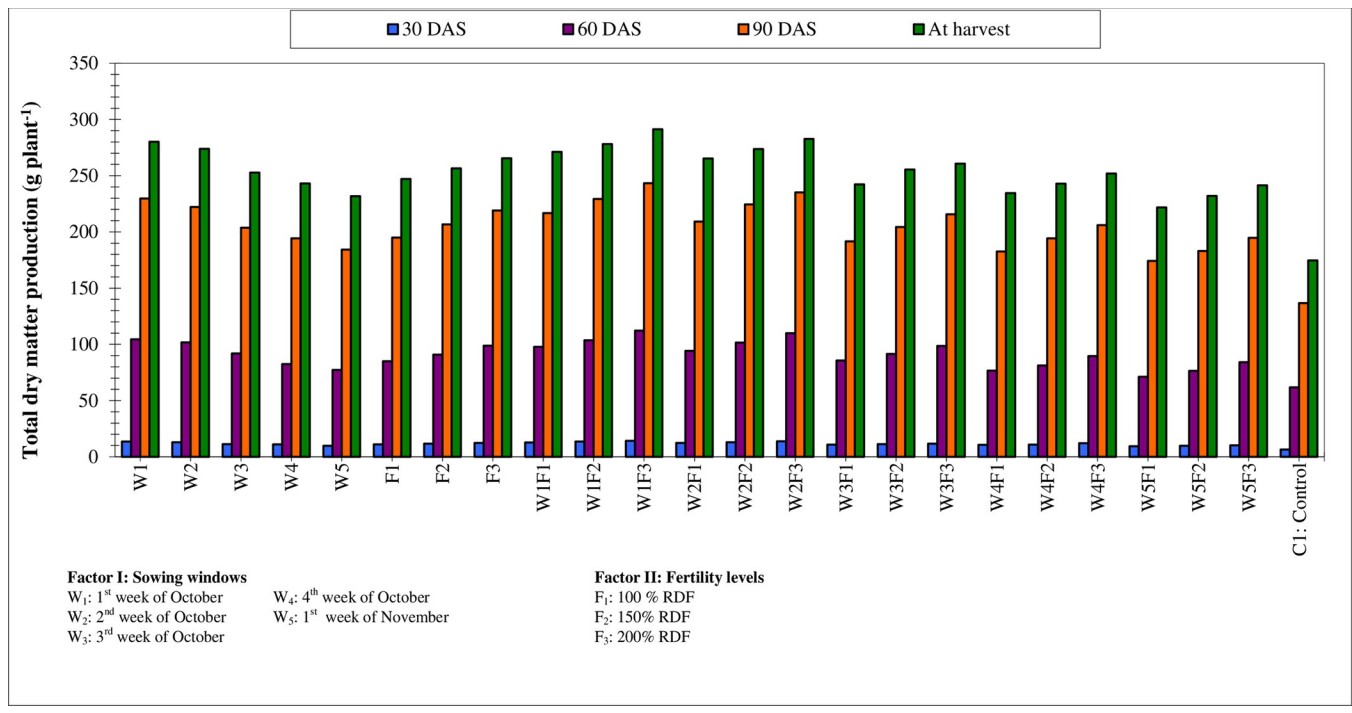

**Fig 3. Total dry matter production at different growth stages of winter maize as influenced by sowing windows and fertility levels.**

planting would lead to a lesser row number and fewer grain numbers in the rows of maize [43–46]. The average row number in each cob (15.1 to 17.4) remained significant among the treatments. Sowing during the 1st week of October along with the application of 200% RDF recorded significantly higher cob girth (17.6 cm) and it was on par with sowing during the 2nd week of October along with the application of either 200% RDF or 150% RDF (17.2 cm and 16.5 cm respectively). Whereas, sowing during the 1st week of November along with the application of 100% RDF recorded lower cob girth (12.1 cm). Similarly, sowing during the 1st week of October along with the application of 200% RDF recorded significantly longer cob (17.4 cm) and it was on par with all sowing windows except sowing during the 1st week of November applied with all fertility levels. And hence there were fewer seed rows and many seeds per row due to the reduced girth and length of the cob. Plant densities also have greater influence on growth and yield parameters of maize apart from sowing dates and fertility [47, 48].

Further, the increase in grain yield and yield attributes in the first week of October planting was due to improved growth parameters *viz.*, many green leaves per plant (S3 Table in S1 File), leaf area, leaf area index (S4 Table in S1 File), total dry matter production (S5 Table in S1 File), absolute growth rate (AGR) and crop growth rate (CGR) (S6 Table in S1 File) as a result of a higher accumulation of growing degree days (GDD), photothermal units (PTU) and heliothermal units (HTU) compared to delayed planting (Table 2). Similar results were obtained [49, 50, 53, 82] who reported that early planting of maize recorded higher grain yield compared to other delayed planting due to higher accumulation of GDD, PTU, and HTU. Delayed planting caused shortening of growing degree days (GDDs) accumulation during planting to physiological maturity [33–35, 49–51].

There was optimum climatic condition (maximum mean temperature 27.9°C and minimum mean temperature 19°C in October month) prevailed for crop sown during first and

second week of October (early) planting; while, delayed planting recorded reduced growth in terms of leaf area and dry matter accumulation. Several references [54–56] proved that the optimum temperature for maize germination, vegetative growth, and flowering are 21°C, 32°C, and 25–30°C respectively. In the present study late sown crops experienced lower temperatures (18°C, 23°C, and 23.1°Cduring germination, growth, and tasseling respectively) which affected crop yield. Further lesser availability of solar radiation (PTU) as a result of shorter day lengths in late planting (November) condition leads to a shorter growing period (Table 2) which reduced the vegetative growth, dry matter accumulation (Fig 3A), and finally the yield [55, 56]. Environmental changes associated with different planting windows (sunshine and temperature) have a modifying effect on the growth and development of maize plants [55, 56]. In early planted maize, better photosynthesis was observed as evidenced by more leaf area index and accumulation of photosynthates due to favorable climatic conditions [52, 53]. Late planting brings horse weather parameters such as temperature, solar radiation, humidity during crop season which adversely affect the morphology, plant physiology, and molecular level of plants [54, 59].

## Productivity of winter maize as influenced by fertility levels

Application 200% RDF increased the grain yield by 3.71 and 10.98 percent compared to 150% and 100% RDF respectively (Table 3 and Fig 2). The increased grain yield was due to improved yield attributes. Among several inputs essential for crop production, fertilizer management is of superlative importance. Further improved yield attributes were due to increased leaf area, leaf area index, and total dry matter production (Fig 3B). Steady increase AGR and CGR also play an important role in yield. Increased growth and yield parameters in 200% RDF were also due to higher available nutrients and their uptake [55, 56]. The higher yield and its parameters were reported in higher fertilizer levels (300:105:105 kg N-$P_2O_5$-$K_2O$ ha$^{-1}$) in southern India as the plant could express its full genetic potential and better fertilizer levels reduced the cob barrenness percentage [56, 57]. There was a movement of photosynthates from source to sink and a better physiological process. The improved growth and yield attributes at higher nitrogen and phosphorus levels which was due to congenial nutritional environment for plant system on account of their greater availability from the soil, which resulted in the greater synthesis of amino acids, proteins, and growth-promoting substance, which enhanced the meristematic activity and increased the cell division and cell elongation [58]. Further application of a higher dose of fertilizer has increased interception, absorption, and utilization of radiant energy which in turn increased photosynthesis and thereby plant height, stem girth, and finally dry matter accumulation.

## Productivity of winter maize as influenced by the interaction of planting windows and fertility levels

At a 5 percent level of significance, planting during the 1$^{st}$ week of October along with the application of 200% RDF ($W_1F_3$) recorded significantly higher grain and stover yield (9142 kg ha$^{-1}$ and 13050kg ha$^{-1}$ respectively) and it was on par with both planting during 2$^{nd}$ week of October along with the application of 200% RDF ($W_2F_3$) and planting during the 1$^{st}$ week of October along with the application of 150% RDF ($W_1F_2$) (Table 3, Fig 2). The increase in yield and yield attributes was due to higher growth in terms of green leaves per plant, leaf area, leaf area index, and total dry matter production (Fig 3) which increased AGR and CGR. Early planting and supplied and higher fertilizer levels favor good plant height, leaf area index, and dry weight per plant due to favorable climatic conditions especially temperature which increased metabolic activities, increased assimilation, and cell division within the plant [59]. Increased growth attributes at $W_1F_3$ were due to a higher accumulation of GDD, PTU, and HTU (Table 2). However, other growth resources are kept uniform (local control) for all

treatment combinations. The increased application of nutrients increases the uptake of nutrients by plants in winter maize which might be due to the congenial nutrient environment in soil and the availability of higher nutrients in the rhizosphere [57, 60].

## Nutrients use efficiency (NUE) as influenced by planting windows

Nutrients use efficiency (NUE) shows the ability of crops to take up and utilize nutrients for maximum yields. NUE depends on the plant's ability to take up nutrients efficiently from the soil but also depends on internal transport, storage, and remobilization of nutrients. NUE of applied fertilizers is very low due to many reasons like surface runoff, leaching, volatilization, denitrification, and fixation in the soil. The increased yield levels show the higher nutrient use efficiency. The better planting date will provide a congenial environment for plants to uptake more nutrients so that the productivity of crops is increased.

Significantly higher $AE_N$, $AE_P$, and $AE_K$ were recorded from crop sown during $1^{st}$ week of October and lower from crop sown during $1^{st}$ week of November (Table 4) due to higher grain yield in the first planting compared to yield from delayed planting. Similarly, higher $RE_N$, $RE_P$, and $RE_K$ were recorded from crop sown during $1^{st}$ week of October (Table 4) due to higher nutrient uptake. And lower recovery efficiency was obtained from crop sown during $1^{st}$ week of November on the contrary the highest PE was recorded when the crop was sown during $1^{st}$ week of November for nitrogen, phosphorus, and potassium and was on par with crop sown during $1^{st}$ week of October (Table 4). This indicates more capacity of the plant to increase yield with per unit nutrient uptake [61, 62].

## Nutrient use efficiency (NUE) as influenced by fertility levels

Significantly higher $AE_N$, $AE_P$, and $AE_K$ were recorded with the application of 100% RDF (Table 4). On the contrary, lower agronomic efficiency for nutrients was recorded with higher fertility levels. For nitrogen, similar results were noticed by Vanlauwe et al. [63] in a maize-based system and according to them, higher agronomic efficiency was recorded in lower nitrogen levels. Similar results were also obtained by Caviglia et al. [64] who concluded higher agronomic efficiency in lower fertilizer levels in both early and late sown maize. Similarly higher $PE_N$, $PE_P$, and $PE_K$ were recorded with the application of 100% RDF and lower physiological efficiency was recorded with higher fertility levels (Table 4). Similarly, the higher $RE_N$, $RE_P$, and $RE_K$ were obtained with the application of 100% RDF, whereas lower recovery efficiency was observed in higher fertility levels (Table 4). Lesser the application of fertilizer higher will be the nutrient use efficiency [65–68]. This result is also in conformity with the findings of Choudhary et al. [66] and they concluded that yield increase was decreased with each increased level of nitrogen application. The highest agronomic nitrogen use efficiency was recorded with 60 kg N/ha. N level of 180 kg/ha was recorded the least. Yield increase due to per unit increase in uptake of N was decreased with increased levels of N application. The highest NUE always occurs at the lower parts of the yield response curve, where fertilizer inputs are the lowest. The effectiveness of fertilizers in increasing crop yields and optimizing farmer profitability should not be sacrificed for the sake of efficiency alone. There must be a balance between optimum NUE and optimal crop productivity [67].

Increased levels of fertilizer tend to lower production efficiency. Apparent recovery, which indicates the efficiency of absorption of applied nutrients, decreased at higher levels of fertilizer application. Each crop is having a definite capacity to absorb a certain amount of nutrients, beyond which nutrients could not be taken up by the plants. When a limited quantity of nutrients was applied, the crop can efficiently absorb the available nutrients in the soil solution thereby reducing the nutrient losses and increasing the NUE [68–70].

## Nutrient use efficiency as influenced by interaction of sowing windows fertility levels

Interaction effect showed the higher agronomic efficiency with planting during the 1st week of October along with the application of 100% RDF. The higher recovery efficiency for nitrogen, phosphorus, and potassium was recorded with planting during 1st week of October along with the application of 100% RDF. Whereas, higher physiological efficiency was recorded with planting during 1st week of November along with the application of 100% RDF (Table 4). The higher agronomic efficiency in October 1st week planting along with 100% RDF was due to higher grain yield in early planting and lower fertility level which increased the efficiency. The recovery efficiency was also higher in October 1st week planting which was due to higher applied nutrient uptake. The higher physiological efficiency was recorded during 1st week of November which might be due to more capacity of the plant to increase yield with per unit nutrient uptake. The lesser the application of fertilizer, the higher will be the nutrient use efficiency [5, 61, 71–76].

**Energetic as influenced by planting windows.** In agriculture development, the energy audit of various resources plays a key role in resource management. The changing global climatic conditions and increasingly growing, energy demands necessitate the development of a production system that utilizes less energy and produces more energy as output. The energetics was calculated per hectare and then these input data were multiplied with the conversion factor of its energy equivalent. The energy indices were determined by using standard equations [35].

The total input energy was lower for early planting windows due to lower irrigation requirement [74]. The productivity of the crop sown on the 1st and 2nd week of October was higher than the delayed sown crop which resulted in higher output energy and lower output energy was recorded with crop sown during the 1st week of November (Table 5). The net energy was also higher from early sown winter maize because of higher output energy during these planting windows and lower net energy was recorded with crop sown during the 1st week of November. Further, energy use efficiency was highest with crop sown during the 1st week of October and was found on par with planting during the 2nd week of October and the lowest energy use efficiency was recorded in the last planting during the 1st week of November (Fig 4A). The higher energy use efficiency in early sown maize compared to late planting was due to higher grain yield and output of energy. Energy productivity was also higher from crop sown during the 1st week of October and was on par with planting during the 2nd week of October. Lower energy productivity was observed in 1st week of November planting. Higher energy productivity was directly correlated with higher productivity. The specific energy was higher in 1st week of November planting and lower specific energy was recorded in 1st week of October planting (Fig 4A). The higher specific energy in delayed planting was due to higher energy requirements to produce unit yield. The same results were observed by Puniya et al. [78]. The energy use efficiency (EUE) was significantly positively correlated with net energy return, energy productivity, energy intensity, energy output, helio-thermal use efficiency, heat use efficiency, and significantly negatively correlated with specific energy and helio-thermal units [77, 78].

## Energetics as influenced by fertility levels

According to many researchers, the inputs such as fuel, electricity, machinery, seed, fertilizer, and chemical take a significant share of the energy supplies to the production system in modern agriculture [16]. Foremost important among them is careful management of fertilizers, because on the one hand, in many cases it alone shares more than 50 percent of total input

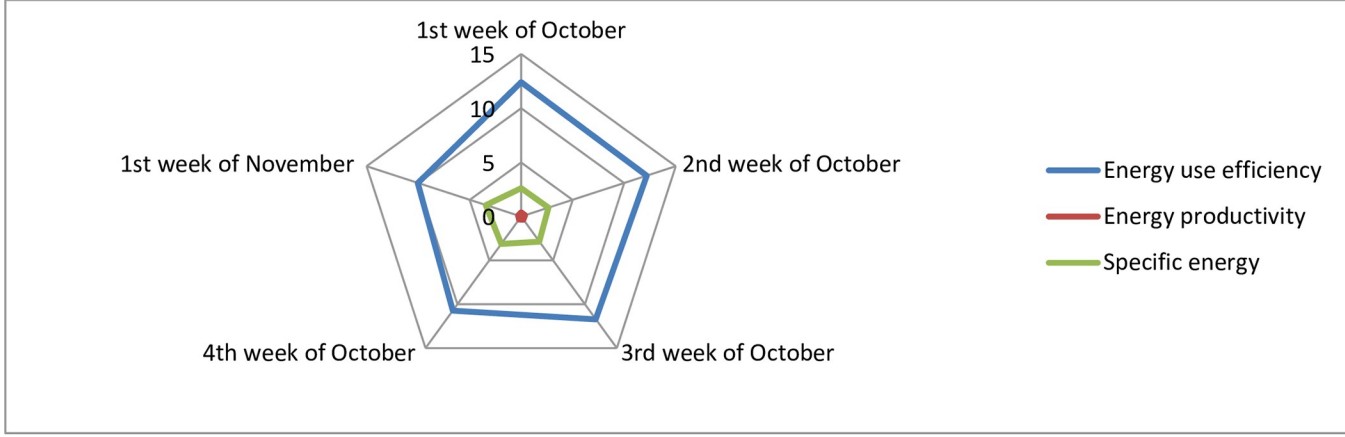

a.

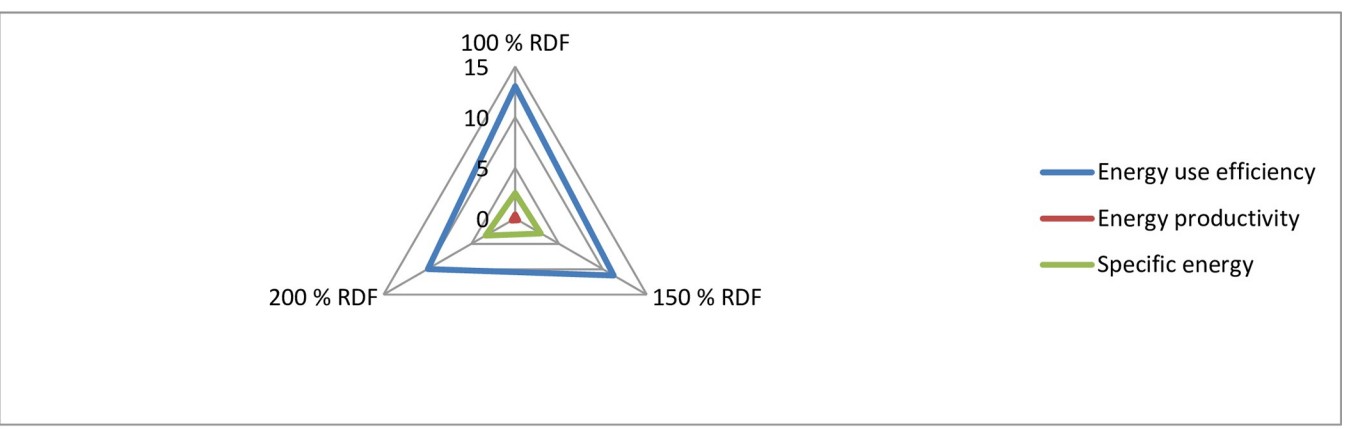

b.

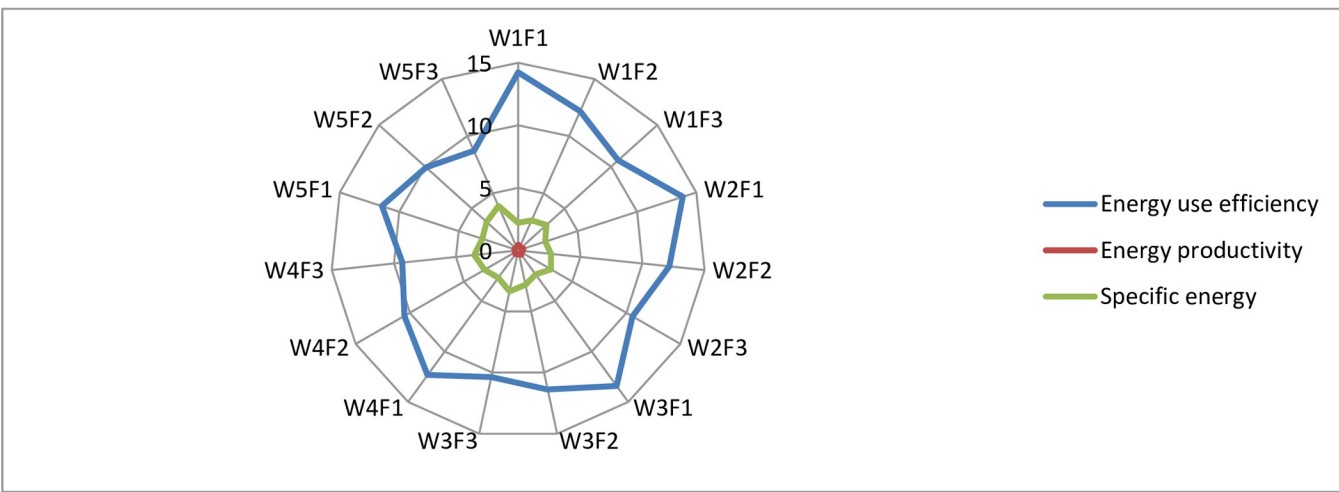

c.

**Fig 4.** (a–c) represent radar chart representing multi-criteria assessment for energy use efficiency, energy productivity and specific energy for different sowing windows, fertility levels and their interactions respectively.

energy used in a system, and the other, it is the most imperative growth factor for proper growth and development of plants [16]. It was observed that the fertilizers had the highest rate of energy equivalency of all the inputs used in maize production at 51.5 percent [6].

The total energy input in 200% RDF was higher than other fertility levels due to the higher rate of application. Aakash et al. [14] reported that fertilizer management is very essential since it utilized almost 70% of the total input energy used in maize production. Application of 200% RDF recorded higher total output energy. The lower total output of energy was recorded with the application of 100% RDF due to lower grain yield. Significantly higher grain and stover yield in higher fertility levels increased the total output energy. Hence, the net energy was higher in 200% RDF, and lower net energy was recorded with the application of 100% RDF. While the energy use efficiency and energy productivity were higher in 100% RDF (Fig 4B). The higher energy use efficiency was due to the higher ratio of output to input energy. Similarly, higher energy productivity was due to a higher ratio of yield to input energy. The specific energy was higher with the application of 200% RDF (Fig 4B). This was due to the higher energy requirement to produce a unit yield in 200% RDF [79–82]. The findings of Khokhar et al. [7] are similar to the above results and they concluded higher input energy, output energy, and energy balance in higher fertility levels and higher energy use efficiency and energy productivity in lower fertility levels in both maize and wheat crops. Singh et al. [67] reported similar results and they reported higher output energy and net energy return in site-specific nutrient management compared to farmer practice and RDF due to higher yield levels in precision nutrient management practices. Choudhary et al. [66], Biswasi et al. [83], Jayadeva and Prabhakar shetty [84], also found that higher input energy, output energy, and net energy in higher fertility levels compared to lower fertility levels.

## Interaction effect of planting windows and fertility levels on energetics

Interaction of planting windows and fertility levels plays an important role in energy flow in winter maize. Relatively higher input energy was recorded in 1st week of November planting along with 200% RDF. Planting during the 1st week of October along with application 200% RDF ($W_1F_3$) (Fig 4C) recorded higher total output energy compared to other treatment combinations. The higher output energy was due to higher yield levels in $W_1F_3$. Higher net energy was recorded with planting during the 1st week of October along with the application of 200% RDF and it was found on par with early planting during the 2nd week of October and 1st week of October along with the application of either 200% RDF or 150% RDF. This was because there was higher input energy use which increased the grain and stover yield resulting in increased the total output energy and net energy, whereas higher energy use efficiency was recorded with planting during 1st week of October along with the application of 100% RDF and was found on par with planting during 2nd week of October along with the application of 100% RDF, planting during 3rd week of October along with the application of 100% RDF (Fig 4C). This was because of the higher ratio of output to input energy. Similarly higher energy productivity was recorded with planting during 1st week of October along with the application of 100% RDF and it was on par with planting during the 2nd week of October along with the application of 100% RDF, planting during the 3rd week of October along with the application of 100% RDF (Fig 4C). Similar results were recorded [79, 83]. The higher energy productivity was due to a higher ratio of grain yield to energy input. Higher specific energy was recorded with planting during the 1st week of November along with the application of 200% RDF and was found on par with planting during the 4th week of October along with the application of 200% RDF (Fig 4C). This was because in this treatment combination there was a higher requirement of energy to produce unit yield [78, 79, 84].

## Economics

Significantly higher gross return, the net return, and B-C ratio were recorded with planting during the 1st week of October and it was on par with planting during the 2nd week of October (Table 6). The higher gross return and net return were due to higher grain yield and stover yield in these two planting windows, whereas significantly lower gross return, net return and B-C ratio recorded with planting at 1st week of November were due to lower productivity [46]. Among the fertility levels, significantly higher gross return and net return was recorded with the application of 200% RDF and was on par with the application of 150% RDF due to higher grain and stover yield in these two fertility levels, whereas significantly lower gross return and net return was recorded with the application of 100% RDF due to its lower grain and stover yield. There is no significant difference in the B-C ratio [75, 85]. As both gross return and cost of production are increased at a similar scale/ratio in all the fertility levels and hence there is no significant difference concerning the B-C ratio.

Interaction effect showed that significantly higher gross return and net return were recorded with planting during 1st week of October along with the application of 200% RDF and were on par with planting during 2nd week of October with 200% RDF and planting during 1st week of October with 150% The higher gross return and net return in these interactions were due to higher grain and stover yield, whereas significantly lower gross return and net return were recorded with planting during 1st week of November along with the application of 100% RDF was due to its lower productivity of crop [76]. A significantly higher B-C ratio was recorded with sowing at 1st week of October along with the application of 150% RDF (2.55) and lower was with sowing at 1st week of November along with the application of 100% RDF (1.83).

## Conclusion

Growing maize in the winter season is more suitable than the rainy season by looking at crop growth, productivity and profitability. The best planting windows during winter to obtain higher productivity, NUE, energy use efficiency, energy productivity, the net return, and the B-C ratio were 1st and 2nd week October. Among fertility levels, application of either 200%or 150% RDF showed higher productivity. Whereas, higher NUE, energy use efficiency, and energy productivity were recorded with 100% RDF. Considering all these variables studied, it could be inferred that planting maize during either first or second week of October along with application 150% RDF is most productive, remunerative, and energy-efficient in south India.

## Supporting information

**S1 File.**
(DOCX)

## Author Contributions

**Conceptualization:** S. R. Salakinkop.

**Data curation:** S. R. Salakinkop.

**Formal analysis:** Siddharth Hulmani, S. R. Salakinkop.

**Investigation:** S. R. Salakinkop.

**Methodology:** S. R. Salakinkop.

**Project administration:** S. R. Salakinkop.

**Resources:** S. R. Salakinkop, G. Somangouda.

**Supervision:** S. R. Salakinkop, G. Somangouda.

**Validation:** S. R. Salakinkop.

**Writing – original draft:** Siddharth Hulmani, S. R. Salakinkop.

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
