## [Decision Letter · Decision Letter 0]

13 Jan 2022

PONE-D-21-37373Productivity, nutrient use efficiency, energetics and bio-economics of winter maize in south IndiaPLOS ONE

Dear Dr. Salakinkop,

Thank you for submitting your manuscript to PLOS ONE. After careful consideration, we feel that it has merit but does not fully meet PLOS ONE’s publication criteria as it currently stands. Therefore, we invite you to submit a revised version of the manuscript that addresses the points raised during the review process.

The manuscript has been reviewed and the two reviewers are in agreement that the manuscript needs major revision. Please follow the comments provided by both authors to prepare your revisions.

We look forward to receiving your revised manuscript.

Kind regards,

Paulo H. Pagliari

Academic Editor

PLOS ONE

Journal Requirements:

5. Please upload a copy of Figure 4, to which you refer in your text on page 8. If the figure is no longer to be included as part of the submission please remove all reference to it within the text.

Reviewers' comments:

Reviewer's Responses to Questions

**Comments to the Author**

1. Is the manuscript technically sound, and do the data support the conclusions?

Reviewer #1: Partly

Reviewer #2: Partly

2. Has the statistical analysis been performed appropriately and rigorously? 

Reviewer #1: Yes

Reviewer #2: Yes

3. Have the authors made all data underlying the findings in their manuscript fully available?

Reviewer #1: Yes

Reviewer #2: Yes

4. Is the manuscript presented in an intelligible fashion and written in standard English?

Reviewer #1: No

Reviewer #2: Yes

5. Review Comments to the Author

Reviewer #1: Review report to the authors of the manuscript PONE-D-21-37373, entitled "Productivity, nutrient use efficiency, energetics and bio-economics of winter maize in south India".

The study seeks to determine the best sowing periods for winter maize in Southern India using an experiment that was based on a factorial randomized complete block design. The authors used three replications and fifteen treatment conditions in different planting windows and fertility levels as factors in the experiment. The methodology is sound for such analysis, and the paper addresses an interesting issue, especially for improving maize biomass and economic return during the wintry agronomic period in Southern India. The paper is overall well-written, although careful professional English proofreading is necessary.

Below, I present further comments the authors should consider:

Abstract

1. “Maize area is rapidly spreading in south India in response…” Is maize spreading? With the way this was presented, it appears that the increase in the area where maize is cultivated is a natural phenomenon rather than the fact that farmers themselves are the ones cultivating and increasing the farm area dedicated to maize production.

2. “RDF(recommended dose of fertilizer)”. Normally, abbreviations should come after the full meaning is presented. Please reverse the order and include a space between RDF and the bracket.

3. The research gap, objective, methods used and contribution(s) of the study are missing or not clear in the abstract. Please rework your abstract to depict these important aspects of your manuscript.

4. Please re-read and re-edit the whole abstract. Some sentences are incomplete while others are hard to understand. For example, “Also it recorded higher net returns and gross returns Whereas, energy use efficiency and energy productivity were higher with planting during first week of October along with application of 100 % RD”

5. Also, several grammatical, punctuation, and spelling issues were spotted.

Introduction

1. Please provide a reference for this statement of fact: “The world's maize area is 192.50 million hectares, and it ranks first in production with 1,112.40 million metric tonnes.”

2. Replace “it” with maize so that the reader can understand the sentence well: “After rice and wheat, it is India's third most popular crop”

3. “..Because of its photo-thermo-insensitive” => Correct the double full stops. Also, check for similar errors throughout the manuscript.

4. The authors tend to use maize and corn interchangeably in the manuscript. Please note that corn is used in North American English, while maize is used in British English. Moreover, the two terms can also have substantially different applications. I will suggest that you stick to one term, preferably maize since that is used more in the manuscript.

5. “There are three distinct seasons for the cultivation of maize in India viz., Kharif, rabi in peninsular India and Bihar, and spring in northern India. Maize is predominantly a kharif season crop but in past few years winter maize has gained a significant place in total maize production in India[3].” Please rework this sentence because different font types and sizes were used. What do you mean kharif season? It may be good if you provide a short footnote describing that.

6. A whole of the first part of the introduction was dedicated to (winter) maize description, without arguments linking your keywords such as fertility, bioeconomics, energetic, and planting, etc. The authors should focus more on presenting arguments linking broader issues on productivity, nutrient use, energetics or bioeconomy during the winter season.

7. Just like the abstract, the research objective of the study is not clearly stated. Several questions should be answered in the introduction: what is the research or knowledge gap (s) that the study tries to fill, how was the gap(s) resolved and what contribution is the study making to the existing body of literature?

8. “There are a several of other factors that influence winter maize production and productivity; however, fertiliser management is one of the most important factors influencing maize growth and yields. Early maize planting can improve grain yields significantly, but other practises such as fertility can also trigger the yield[7]” => Please provide a reference after “… influencing maize growth and yields” These sentences are not clear. Is it fertilizer management that influences maize growth and yields or just fertilizer? In what direction would the factor influence maize growth and yields?

9. “One of them is fertiliser control with care. Since, on the one hand, it accounts for more than half of the total input energy used in the maize production system in many cases, and on the other hand, it is the most important factor for proper plant growth and development.” More clarification is needed. How? Concerning the total input energy, are the authors referring to on-farm or off-farm activities, or both? Regardless, a reference is needed here.

10. This sentence is incomplete: “As a result, excessive deployment, wastes resources and money while also exacerbating environmental problems[11].”

11. Please present how the remaining part of the manuscript is structured at the end of the introduction.

Materials and Methods

1. If you compare this “was conducted to investigate the response of winter maize to planting windows and fertility levels at the University of Agricultural Sciences, Dharwad (Karnataka)…” with “Hence experiment was conducted to explore the most congenial sowing period in Southern India” in the introduction, the purported objectives of the study are not the same. What is the broad objective of the study, and what specific objectives or questions does the manuscript answer? More specifically, south India was the focus in the title and the objective in the introduction, what is the reason(s) for using samples from the University of Agricultural Sciences? Would you consider your result as being valid and generalizable to other parts of Southern India?

2. “The soil is classified as clay by the USDA soil textural classification table.” => Please provide a reference here. Aside from the soil type, more information about the soil composition should be provided.

3. “Planting windows (1st week of October, 2nd week of October, 3rd week of October, 4th week of October, and 5th week of October) and fertility levels (100 percent recommended dose of fertiliser (RDF), 150 percent RDF, and 200 percent RDF) were used as factors in the experiment.” => Please present the actual dates and the year of the planting windows, instead of just weeks. This will enhance the reliability of the materials presented, as well as the replicability of your study.

4. How many controls were considered in the study?

5. Please first provide the full meanings before using acronyms e.g. for FYM and NUE.

6. “The nutrients viz., nitrogen, phosphorus and potassium were applied @ 150 kg N ha-1, 65 kg P2O5 ha-1 and 65 kg K2O ha-1 for fertility level of 100 % RDF. Similarly for 150 % RDF, 225 kg N , 97.5 kg P2O5 and 97.5 kg K2O ha-1 was applied. And for 200 % RDF, 300 kg N, 130 kg P2O5 and 130 kg K2O ha-1 was applied through urea, diammonium phosphate (DAP) and muriate of potash (MOP) respectively. FeSO4 and ZnSO4 were applied @ of 25 kg ha-1.” => The authors should rework this aspect. It is not fully clear. What does viz stands for here?

7. The authors tend to use @ instead of “at” throughout the manuscript. Please correct this. For example, “FeSO4 and ZnSO4 were applied @ of 25 kg ha-1.”

8. Growing degree day: Please provide more description and justification on the reason(s) this method and the subsequent ones (e.g. Photothermal units, and Heliothermal units, etc.) are important. I assume that there are plausibly other methods that can be employed to attain similar results. Also, the processes involved in the implementation of these methods and their parameters are not clear. For GDD, which dates were the maximum and minimum temperatures reported, and how were these data captured. Providing such details would be good for non-expert readers and would also be crucial in aiding the readers to understand the framework followed in the study.

9. Also, are the formulas presented standardized? If yes, please provide references for them.

10. Physiological efficiency (PE) indicates grain yield increase in kg per kg nutrient uptake from fertilizer[18]. And expressed in kilogram per hectare (kg ha-1). => Please rework this section, it is not clear.

11. Bio-economics: Is this the right term for what you intend to describe? Please note that bioeconomy is characterized by the utilization of biological resources from land and sea (e.g. plant and animal materials) for the production of a wide range of sustainable products and services aimed at driving economic growth through knowledge-based inventions and innovative biotechnological processes.

12. “The price in USD of the inputs prevailed at the time.” It is not clear which inputs you are referring to here. Is the cost of the labor hours for preparing the land, and planting and harvesting also considered? The authors should which input costs were considered to avoid confusion for the readers.

13. “The energy balance was calculated using the data on input energy, output energy.” Please provide further information, this statement is not clear.

14. The equations under Energetics are not well-formatted. Please rework them appropriately

15. “The highest values were denoted by the letter ‘a,' which was followed by the alphabets for lower values (b, c, d, etc.).” => Be precise with the information provided in this sentence. The values of what?

16. The authors should link each method to specific objectives to be answered. Mere presenting the methods is not enough, the rationale for choosing each method should be clear.

Results

1. Subheading: “Weather, GDD, PTU, HTU and days for physiological maturity.” Would a subheading “Descriptive statistics” be more appropriate as the heading here?

2. “During the winter cropping season (2019-2020), a total rainfall of 352.0 mm was received out of which 323.2 mm was received during the planting month (October) (Fig.4). The highest and the lowest maximum temperature were 31.8 ºC (February) and 28.5 ºC (December), respectively…” This information has been presented in the methodology section, so it feels more like repetition.

3. Subheading: “Nutrient use efficiency in winter maize (NUE)” => NUE should be placed in the correct position immediately after efficiency. Rather, the author can do away with the abbreviation since it has been provided earlier in the manuscript

4. “Planting during 1st week of October along with application of 100 % RDF recorded significantly higher AEN, AEP and AEK (29.59, 68.29 and 68.29 kg kg-1 respectively) and it was found on par with planting during 2nd week of October along with application of 100 % and planting during 3rd week of October along with application of 100 % RDF.” => What does par mean in this context? Does it mean that there is no significant difference between the two planting periods when those parameters are compared? If so, please rework this throughout the manuscript as on par does not provide a statistical connotation.

5. “Interaction effect found non-significant with respect to PEP and PEK.” This sentence is not clear

6. Subheading Energetics: “Energetics of winter maize significantly influenced by planting windows and fertility levels.” It is not clear how this result was obtained. This points to my comment 15 on the methodology section. The authors should rework the authors' method section to depict how each objective is resolved and their results obtained.

7. The subheadings in the methods and results sections are virtually the same. Authors should clearly distinguish the subheadings as they might become confusing for the readers to follow. For example, you have Bio-economics in the methods, results and discussion sections. In the results section, “Economic value of produced maize biomass” can be used as a subheading instead.

Discussion

1. “Late planting would lead to a lesser row number and less grain numbers in the rows of maize” How? Further clarification is needed.

2. “There was optimum climatic condition (maximum mean temperature 27.9 oC and minimum mean temperature 19 oC in October month) prevailed for crop sown during first and second week of October (early) planting”. This sentence is not clear.

3. “The increased grain yield was due to improved yield attributes.” => This seems redundant and confusing. How is grain yield different from yield attributes? The authors should provide more explanations here.

4. “Planting during 1st week of October along with application of 200 % RDF (W1F3) recorded significantly higher grain and stover yield” => The authors should remind the readers of the analytical methods employed to obtain the results being discussed. There seem to be some spacing issues here too. How are we sure that the higher grain and Stover yields are coming only from the RDF application? The descriptive analysis made is not sufficient to make such a conclusion.

5. “The energy use efficiency (EUE) was significantly positively correlated with net energy return, energy productivity, energy intensity, energy output, helio-thermal use efficiency, heat use efficiency and significant negatively correlated with specific energy and helio- thermal units” Which of the results indicate this in the results section?

6. “According to many researchers the inputs such as fuel, electricity, machinery, seed, fertilizer and chemical take significant share of the energy supplies to the production system in modern agriculture.” Please provide a reference for those studies.

7. “Among the fertility levels, significantly higher gross return and net return was recorded with application of 200 % RDF and was on par with application of 150 % RDF due to higher grain and stover yield in these two fertility levels, whereas significantly lower gross return and net return was recorded with application of 100 % RDF due to its lower grain and stover yield. There is no significant difference with respect to B-C ratio” => The authors should discuss the reason(s) for such a result.

Discussion

1. The authors should provide caveats other researchers or readers should note in the interpretation of the results and overall implementation of the study.

2. The future direction for research should also be outlined and discussed.

3. The implications of the study particularly for farmers in South India should also be highlighted.

Other general comments

1. Please include page numbers. This is one of the guidelines of the journal: "Include page numbers and line numbers in the manuscript file. Use continuous line numbers (do not restart the numbering on each page)."

2. The headings should also be numbered to allow for easy differentiation of main headings from subheadings.

3. Please, thoroughly check the whole manuscript for double spaces and full-stops.

4. Finally, subject the manuscript to a professional English proofreader and editor.

Reviewer #2: The manuscript is well articulated about optimization of sowing and nutrient management in winter maize. There are some typo and punctuation error in MS which needs to corrected. This needs some important revisions for more clarity:

In abstract: There is need to discuss interaction effects, if any. a line of recommendation needs to be added.

In line 1-5 of introduction there is two full stop (..) at two places, remove one.

In , Andra Pradesh (9.5 %), check spelling of Andhra Pradesh

Exhaustive review needs to be included for identification of the problem for research. Include a para hypothesis clearly at the end of introduction.

In Materials and Methods, 'With a spray of proclaim @ of 0.5 g litre-1', mention here technical name of it and also mention time of spray and how many spray were undertaken.

In Physiological efficiency (PE) formula, mention what A and F indicates.

The bio-economics is new term used. How it is different from economics which is mostly reported as per studied parameter in the study. This needs to be economics in whole study.

In ' Planting during 1st week of October recorded significantly

higher GDD, PTU and HTU accumulation (1530.1 0C day, 17371.3 0C day hr and 11758.5

0C day hr respectively) and further it took significantly more number of days for

physiological maturity (120.0 days).' use insert symbol for mentioning the degree not superscript. Follow it in whole MS.

In 'Delayed planting caused shortening of growing degree days (GDDs)

accumulation during planting to physiological maturity[30]. ' The GDD is fixed, how this can be shortened, number of days can be lesser in delayed planting to increase in temperature may lead to forced maturity. This need to corrected accordingly.

How the agronomic or physiological efficiency of the applied NPK were calculated without any control treatment in the study. This portion needs to be removed and partial factor productivity if studied may be included here.

The Fig. 1 is not a good representation of data. Needs to make as a bar diagram.

The Fig. 3 is also not giving any new information and needs to be dropped as it is not good representation of the data as well.

6. PLOS authors have the option to publish the peer review history of their article (what does this mean?). If published, this will include your full peer review and any attached files.

Reviewer #1: No

Reviewer #2: No

---

## [Author Response · Author response to Decision Letter 0]

22 Feb 2022

Response to Academic Editor and Reviewers' Comments.

Ref: PONE-D-21-37373 and PONE-D-21-37373R .Productivity, nutrient use efficiency, energetics and bio-economics of winter maize in south India

Abstract

1. “Maize area is rapidly spreading in south India in response…” Is maize spreading? With the way this was presented, it appears that the increase in the area where maize is cultivated is a natural phenomenon rather than the fact that farmers themselves are the ones cultivating and increasing the farm area dedicated to maize production.

Answer: Due to absence of major environmental impediments in winter, the desired field and crop operations can be planned and executed at the most desired time. And hence farmers are using this potential option to increase both area and production. Hence there is immense potential to increase the area under winter maize cultivation (line no 13-14).

2. “RDF(recommended dose of fertilizer)”. Normally, abbreviations should come after the full meaning is presented. Please reverse the order and include a space between RDF and the bracket.

Answer: Suggestion is incorporated in following sentence=> The present investigation encompassing different sowing windows with different fertility levels revealed that significantly higher winter maize productivity was achieved from either first and second week of October planting along with application of 200 % recommended dose of fertilizer (RDF) followed by 150 % RDF (line no 24).

3. The research gap, objective, methods used and contribution(s) of the study are missing or not clear in the abstract. Please rework your abstract to depict these important aspects of your manuscript.

4. Answer: Abstract has been revised by including research gap, objectives, methods used and contribution. The important information being added is as follow=> There were no planned field experiments to explore and optimize right time of sowing and quantity of fertilizer to be added previously due to presence of negligible winter maize area. Farmers’ used to cultivate maize as per their choice of sowing time with application of quantity of fertilizer recommended for rainy season maize. There was no efforts made towards working of economic analysis including energy budgeting. And hence the investigation was conducted with objective to explore the optimal planting period and fertilizer levels for winter maize through economic and energy budgeting. Planting windows (1st week of October, 2nd week of October, 3rd week of October, 4th week of October, and 5th week of October) and fertility levels (100 percent recommended dose of fertilizer (RDF), 150 percent RDF, and 200 percent RDF) were used as factors in Factorial Randomized Complete Block Design (RCBD) with three replications (line no 12-26 and 43-46).

.

5. 4. Please re-read and re-edit the whole abstract. Some sentences are incomplete while others are hard to understand. For example, “Also it recorded higher net returns and gross returns Whereas, energy use efficiency and energy productivity were higher with planting during first week of October along with application of 100 % RD”

Answer: Entire abstract has been re-edited taking into consideration of all suggestions (line no 12-26 and 43-46).

Introduction

1. Please provide a reference for this statement of fact: “The world's maize area is 192.50 million hectares, and it ranks first in production with 1,112.40 million metric tonnes.”

6. Answer: Reference is already quoted both in text and reference as =>[1] (line no 52).

2. Replace “it” with maize so that the reader can understand the sentence well: “After rice and wheat, it is India's third most popular crop”

Answer: Replaced as => After rice and wheat, maize is India's third most popular crop(line no 53).

3. “..Because of its photo-thermo-insensitive” => Correct the double full stops. Also, check for similar errors throughout the manuscript.

Answer: Double full stops replaced with single and similar mistakes were checked in entire text.

4. The authors tend to use maize and corn interchangeably in the manuscript. Please note that corn is used in North American English, while maize is used in British English. Moreover, the two terms can also have substantially different applications. I will suggest that you stick to one term, preferably maize since that is used more in the manuscript.

Answer: only one term “Maize” is used in entire text. 

5. “There are three distinct seasons for the cultivation of maize in India viz., Kharif, rabi in peninsular India and Bihar, and spring in northern India. Maize is predominantly a kharif season crop but in past few years winter maize has gained a significant place in total maize production in India[3].” Please rework this sentence because different font types and sizes were used. What do you mean kharif season? It may be good if you provide a short footnote describing that.

Answer: different font types and sizes were replaced with same font and size matching to remaining text. Kharif season is replaced with rainy season and rabi season is replace with winter season in entire text.

6. A whole of the first part of the introduction was dedicated to (winter) maize description, without arguments linking your keywords such as fertility, bioeconomics, energetic, and planting, etc. The authors should focus more on presenting arguments linking broader issues on productivity, nutrient use, energetics or bioeconomy during the winter season.

Answer: Following points were added to first part of existing introduction=>.

Nutrients use efficiency (NUE) shows the ability of crops to take up and utilize nutrients for higher productivity [4,58]. NUE depends on the plant’s ability to take up nutrients efficiently from the soil, but also depends on internal transport, storage and remobilization of nutrients. NUE of applied fertilizers may very low due to many reasons like surface runoff, leaching, volatilization, denitrification and fixation in the soil. The increased productivity show the higher nutrient use efficiency. The better planting date will provide the congenial environment to plants to uptake more nutrients so that productivity of crops is increased. In agriculture development, the energy audit of various resources plays a key role in resource management. Under the changing global climatic conditions and increasingly growing energy demands necessitate the development of a production system which utilizes less energy and produces more energy as output [4]. Fertilizer as input had the highest rate of energy equivalency of all the inputs used in maize production at 51.5 per cent[12] . The findings of Khokhar et al.[7] recorded higher input energy, output energy and energy balance in higher fertility levels and higher energy use efficiency and energy productivity in lower fertility levels (line no 72-87, 99-101). 

 In both rainy and winter season, higher gross return and net returns were obtained from maize sown early in the season compared to late sown crop [8- 9] (line no 84-87).

7. Just like the abstract, the research objective of the study is not clearly stated. Several questions should be answered in the introduction: what is the research or knowledge gap (s) that the study tries to fill, how was the gap(s) resolved and what contribution is the study making to the existing body of literature?

Answer: research objective, knowledge gap (s) and contribution is the study making to the existing body of literature was added at the end to introduction (line no 121-127). 

8. “There are a several of other factors that influence winter maize production and productivity; however, fertiliser management is one of the most important factors influencing maize growth and yields. Early maize planting can improve grain yields significantly, but other practises such as fertility can also trigger the yield[12]” => Please provide a reference after “… influencing maize growth and yields” These sentences are not clear. Is it fertilizer management that influences maize growth and yields or just fertilizer? In what direction would the factor influence maize growth and yields?

Answer: Reference is provided as directed > There are a several of other factors that influence winter maize production and productivity; however, fertiliser management is one of the most important factors influencing maize growth and yields [12]. Fertilizer management or recommendations on time and method of application of fertilizers have been already existed and it is similar to rainy season maize. However rate of application need to be worked out in order to meet higher winter maize productivity(line no 99-101).

9. “One of them is fertiliser control with care. Since, on the one hand, it accounts for more than half of the total input energy used in the maize production system in many cases, and on the other hand, it is the most important factor for proper plant growth and development.” More clarification is needed. How? Concerning the total input energy, are the authors referring to on-farm or off-farm activities, or both? Regardless, a reference is needed here.

Answer: First sentence clearly indicate that crop demand- driven approach is followed so that there is no huger for nutrients for showing its full yield potential. And at the same time applied fertilizer energy usage and efficiency were also worked out in order to economize the winter maize productivity. Both on-farm and off-farm activities were considered for working out total input energy referecnces cited are 6,15-16 (line no 108-118).

10. This sentence is incomplete: “As a result, excessive deployment, wastes resources and money while also exacerbating environmental problems[16].”

Answer: Same sentence is re-written> Excessive application of fertilizer results in wastes of resources and money while also exacerbating environmental problems [16] (line no 117-118).

11. Please present how the remaining part of the manuscript is structured at the end of the introduction.

Answer: After inclusion of all suggestion, the introduction flow is as follow.

a. Area, production and distribution of maize in world and India.

b. Importance of maize as feed,food and industry in-put

c. Winter maize productivity and area spread

d. Scope for increasing productivity of winter maize

e. Energy flow as input and output

f. Importance and quantity of fertilizer input requirement

g. Importance of sowing windows in winter

Materials and Methods

1. If you compare this “was conducted to investigate the response of winter maize to planting windows and fertility levels at the University of Agricultural Sciences, Dharwad (Karnataka)…” with “Hence experiment was conducted to explore the most congenial sowing period in Southern India” in the introduction, the purported objectives of the study are not the same. What is the broad objective of the study, and what specific objectives or questions does the manuscript answer? More specifically, south India was the focus in the title and the objective in the introduction, what is the reason(s) for using samples from the University of Agricultural Sciences? Would you consider your result as being valid and generalizable to other parts of Southern India?

Answer: Rainfall and temperature over the locations in south India vary during rainy season. However during winter, there is a need to raise the crop only by providing irrigation in entire maize growing areas’ of southern India. Further winter maize area in N-E India is also increasing and there productivity, crop duration and climate are different from that prevail in maize growing area of south India. Further Indian Institute of Maize Research (IIMR), apex body on maize research grouped entire maize growing area of south India in one zone (Zone-4- peninsular India). And hence to differentiate the regions, the south India is included in the title. General objective was to explore best sowing time and fertility levels. Specific objective were to study energitics and economics including winter maize productivity. These general and specific objectives been achieved for one location by taking into consideration of data of experiment from one location. I feel these results may stand base for studying variability of optimum sowing time and fertility across south India.

2. “The soil is classified as clay by the USDA soil textural classification table.” => Please provide a reference here. Aside from the soil type, more information about the soil composition should be provided.

Answer: The soil is classified as clay by the USDA soil textural classification table [17]. Soil composed of course sand, fine sand,silt and clay by 6.23,12.66,28.17 and 52.93 respectively(line no 135-137).

3. “Planting windows (1st week of October, 2nd week of October, 3rd week of October, 4th week of October, and 5th week of October) and fertility levels (100 percent recommended dose of fertiliser (RDF), 150 percent RDF, and 200 percent RDF) were used as factors in the experiment.” => Please present the actual dates and the year of the planting windows, instead of just weeks. This will enhance the reliability of the materials presented, as well as the replicability of your study.

Answer: First to fifth dates of sowing were 05-10-2019, 12-10-2019, 18-10-2019, 28-10-2019 and 06-11-2019 respectively. It is included the text (line no 142-144).

4. How many controls were considered in the study?

Answer: Interaction of November sowing with application of 100 per cent RDF is considered as control (line no 144-146)

5. Please first provide the full meanings before using acronyms e.g. for FYM and NUE.

Answer: Full meanings before acronyms are provided in entire text.

6. “The nutrients viz., nitrogen, phosphorus and potassium were applied @ 150 kg N ha-1, 65 kg P2O5 ha-1 and 65 kg K2O ha-1 for fertility level of 100 % RDF. Similarly for 150 % RDF, 225 kg N , 97.5 kg P2O5 and 97.5 kg K2O ha-1 was applied. And for 200 % RDF, 300 kg N, 130 kg P2O5 and 130 kg K2O ha-1 was applied through urea, diammonium phosphate (DAP) and muriate of potash (MOP) respectively. FeSO4 and ZnSO4 were applied @ of 25 kg ha-1.” => The authors should rework this aspect. It is not fully clear. What does viz stands for here?

Answer: Prefix The nutrients viz., is deleted and sentence has started with= > Nitrogen(line no 155)

After including your suggestion the sentence are redrafted as > All other treatment plots, including control plots, had well decomposed farm yard manure (FYM) @ 10 t ha-1 incorporated into soil two weeks prior to planting. The nitrogen, phosphorus and potassium were applied @ 150 kg N ha-1, 65 kg P2O5 ha-1 and 65 kg K2O ha-1 for fertility level of 100 % RDF. Similarly for 150 % RDF, 225 kg N, 97.5 kg P2O5 and 97.5 kg K2O ha-1 was applied. And for 200 % RDF, 300 kg N, 130 kg P2O5 and 130 kg K2O ha-1 was applied. Urea, diammonium phosphate (DAP) and muriate of potash (MOP) were used as a source of nitrogen, phosphorus and potassium respectively. Zinc and iron were applied in the form of FeSO4 and ZnSO4 at 25 kg ha-1. 

7. The authors tend to use @ instead of “at” throughout the manuscript. Please correct this. For example, “FeSO4 and ZnSO4 were applied @ of 25 kg ha-1.”

Answer: In entire text ‘@’ is replaced with ‘at’ in entire text.

8. Growing degree day: Please provide more description and justification on the reason(s) this method and the subsequent ones (e.g. Photothermal units, and Heliothermal units, etc.) are important. I assume that there are plausibly other methods that can be employed to attain similar results. Also, the processes involved in the implementation of these methods and their parameters are not clear. For GDD, which dates were the maximum and minimum temperatures reported, and how were these data captured. Providing such details would be good for non-expert readers and would also be crucial in aiding the readers to understand the framework followed in the study.

Answer: Many scientist used these GDD, PTU and HTU as a indicator of influence temperature and radiation on crop performance. There are several references which justify present results and I quate here some examples=> Swetha (2017) reported that early sowing of maize recorded higher grain yield compared to other delayed sowing due to higher accumulation of GDD, PTU and HTU. Sutton and Stucker (1974) reported that delayed sowing causes shortening of growing degree days (GDDs) accumulation during planting to physiological maturity. Similarly higher GDD accumulation was reported in early planted maize (Hugar, 2015). GDD, PTU and HTU units can be used to assess the suitability of a region for production of a particular crop, estimate the growth-stages of crops, predict maturity,best timing of fertilizer or pesticide application; estimate the heat stress on crops; plan spacing of planting dates to produce separate harvest dates (line no 173-178).

9. Also, are the formulas presented standardized? If yes, please provide references for them.

Answer: Formulas presented are referred universally and references for each formulae is already present (reference are[18-20]

10. Physiological efficiency (PE) indicates grain yield increase in kg per kg nutrient uptake from fertilizer[18]. And expressed in kilogram per hectare (kg ha-1). => Please rework this section, it is not clear.

Answer: Yes, rightly pointed out. And it is corrected as=>PE expression is grain yield in kg produced per kg of nutrient taken up from soil and expressed in kg kg-1 (line no 212)

11. Bio-economics: Is this the right term for what you intend to describe? Please note that bioeconomy is characterized by the utilization of biological resources from land and sea (e.g. plant and animal materials) for the production of a wide range of sustainable products and services aimed at driving economic growth through knowledge-based inventions and innovative biotechnological processes (line no 216 and 1).

Answer: Since we are dealing with biological entity (crop) and its products, the term bio-economics was used. And as per your suggestion it has been replaced with ‘economics’ in entire section.

12. “The price in USD of the inputs prevailed at the time.” It is not clear which inputs you are referring to here. Is the cost of the labor hours for preparing the land, and planting and harvesting also considered? The authors should which input costs were considered to avoid confusion for the readers.

Answer: All the inputs=> Land preparation, intercultivation, all applied fertilizers, FYM, seed, plant protection chemicals, irrigation, men and women wages right from sowing to harvesting , drying, processing and marketing of produce were included for working out cost of inputs(line no 118-121) 

13. “The energy balance was calculated using the data on input energy, output energy.” Please provide further information, this statement is not clear.

Answer: The energy balance in terms of net energy, energy use efficiecncy, energy productivity and specific energy was calculated using the data on input energy, output energy using the formula(line no 233-135)

14. The equations under Energetics are not well-formatted. Please rework them appropriately

Answer: Formatting will be done at copy editing stage since in MS wards, they tend to move. 

15. “The highest values were denoted by the letter ‘a,' which was followed by the alphabets for lower values (b, c, d, etc.).” => Be precise with the information provided in this sentence. The values of what?

Answer: Sentence is re-written as=> The highest mean values of all the crop and nutrient parameters statistically analyzed were denoted by the letter ‘a,' which was followed by the next alphabets for lower values (b, c, d etc.). At the 0.05 level of significance, mean values denoted by the same small letter in the column do not vary significantly (line no 254-258).

16. The authors should link each method to specific objectives to be answered. Mere presenting the methods is not enough, the rationale for choosing each method should be clear.

Linking of each method to specific objective has been done at appropriate places as =>. various thermal indices including growing degree days (GDD), Photothermal index (PTI) and heat use efficiency (HUE) for maize were calculated by using standard methods to know their influence on maize productivity. Similarly agronomic efficiency, physiological efficiency and recover efficiency were worked out to know relation between maize yield in response to nutrients applied. The energy balance in terms of net energy, energy use efficiecncy, energy productivity and specific energy were calculated taking into consideration of flow of inputs used for production in each treatment and out put obtained. Gross return, net return and B-C ratio were worked out to know the economics of maize production in each treatment ((line no 173-178, 207-209, 218-229).

 Results

1. Subheading: “Weather, GDD, PTU, HTU and days for physiological maturity.” Would a subheading “Descriptive statistics” be more appropriate as the heading here?

Answer: Subheading is changed as Descriptive statistics(line no 261) 

2. “During the winter cropping season (2019-2020), a total rainfall of 352.0 mm was received out of which 323.2 mm was received during the planting month (October) (Fig.4). The highest and the lowest maximum temperature were 31.8 ºC (February) and 28.5 ºC (December), respectively…” This information has been presented in the methodology section, so it feels more like repetition.

Answer: Yes and deleted from results section

3. Subheading: “Nutrient use efficiency in winter maize (NUE)” => NUE should be placed in the correct position immediately after efficiency. Rather, the author can do away with the abbreviation since it has been provided earlier in the manuscript

Answer: corrected as> Nutrient use efficiency (NUE) in winter maize (line no 312)

4. “Planting during 1st week of October along with application of 100 % RDF recorded significantly higher AEN, AEP and AEK (29.59, 68.29 and 68.29 kg kg-1 respectively) and it was found on par with planting during 2nd week of October along with application of 100 % and planting during 3rd week of October along with application of 100 % RDF.” => What does par mean in this context? Does it mean that there is no significant difference between the two planting periods when those parameters are compared? If so, please rework this throughout the manuscript as on par does not provide a statistical connotation.

Answer: It means that there is no significant difference between the two planting periods when those parameters (AEN, AEP and AEK) are compared. Your views are considered, however it is with respect to NUE and not with other attributes studied (line no.313-323)

5. “Interaction effect found non-significant with respect to PEP and PEK.” This sentence is not clear

Answer: Sentence is redrafted as> Whereas, interaction of sowing window and fertility level effect remained non-significant with respect to both PEP and PEK (line no 544-546).

6. Subheading Energetics: “Energetics of winter maize significantly influenced by planting windows and fertility levels.” It is not clear how this result was obtained. This points to my comment 15 on the methodology section. The authors should rework the authors' method section to depict how each objective is resolved and their results obtained.

Answer: Energetics and energy balance here are a general term and specifically they include variour components such as net energy, energy use efficiecncy, energy productivity and specific energy . And each was explained accordingly as per statistical analysis (line no553-632).

7. The subheadings in the methods and results sections are virtually the same. Authors should clearly distinguish the subheadings as they might become confusing for the readers to follow. For example, you have Bio-economics in the methods, results and discussion sections. In the results section, “Economic value of produced maize biomass” can be used as a subheading instead.

Answer: Suggestion is incorporated in result section

Discussion

1. “Late planting would lead to a lesser row number and less grain numbers in the rows of maize” How? Further clarification is needed.

Answer: It is just one of the references [38] to support low yield in delayed sowing. In my study average row number in each cob (15.1 to 17.4) remained significant among the treatments due to reduced cob girth. Sowing during 1st week of October along with application of 200 % RDF recorded significantly higher cob girth (17.6 cm) and it was on par with sowing during 2nd week of October along with application of either 200 % RDF or 150 % RDF (17.2 cm and 16.5 cm respectively).Whereas, sowing during 1st week of November along with application of 100 % RDF recorded lower cob girth (12.1 cm). Similarly, sowing during 1st week of October along with application of 200 % RDF recorded significantly longer cob (17.4 cm) and it was on par with all sowing windows except sowing during 1st week of November applied with all fertility levels. And hence there was less number seeds per seed row due reduced length of cob and seed size in terms of test weight. Data on these yield attributes will be presented, only after getting your suggestion. Above information is added (line no 416-427). 

2. “There was optimum climatic condition (maximum mean temperature 27.9 oC and minimum mean temperature 19 oC in October month) prevailed for crop sown during first and second week of October (early) planting”. This sentence is not clear.

Answer: It is clearly rewritten as => several references [41,42,43] which proved that optimum temperature for maize germination, vegetative growth and flowering are 21 oC, 32 oC and 25-30 oC respectively. In present study late sown crop experienced lower temperature (18 oC, 23 oC and 23.1 oC during germination, growth and tasseling respectively) which affected crop yield. Further lesser availability of solar radiation (PTU) as a result of shorter day lengths in late planting (November) condition leads shorter growing period (Table 2) which reduced the vegetative growth, dry matter accumulation (Fig. 1a) and finally the yield[42,43]. (Line no 442-448)

3. “The increased grain yield was due to improved yield attributes.” => This seems redundant and confusing. How is grain yield different from yield attributes? The authors should provide more explanations here.

Answer: Now, only one independent variable viz, grain yield has been mentioned instead of both grain yield and yield attributes. Generally all plant parameters including growth, yield parameters and yield are dependent variables for growth resources such as sowing time, fertility, variety, management and climatic etc variables (independent variables). And hence efforts have been made to explain magnitude of variation in independent variables (yield parameters and yield) as influenced by sowing windows and fertility levels (line no.463-466).

4. “Planting during 1st week of October along with application of 200 % RDF (W1F3) recorded significantly higher grain and stover yield” => The authors should remind the readers of the analytical methods employed to obtain the results being discussed. There seem to be some spacing issues here too. How are we sure that the higher grain and Stover yields are coming only from the RDF application? The descriptive analysis made is not sufficient to make such a conclusion.

Answer: At 5 per cent level of significance, planting during 1st week of October along with application of 200 % RDF (W1F3) recorded significantly higher grain and stover yield”. 

Principles of field experimentation such as randomization, replication and local control are followed while imposing treatments (on sowing window and fertility levels). Grain and stover yields are coming not only from the RDF application alone but also along with sowing window. However other growth resources are kept uniform (local control) for all treatment combinations ( line no 481).

5. “The energy use efficiency (EUE) was significantly positively correlated with net energy return, energy productivity, energy intensity, energy output, helio-thermal use efficiency, heat use efficiency and significant negatively correlated with specific energy and helio- thermal units” Which of the results indicate this in the results section?

Answer: It is just the references [74-75] to show universally that energy use efficiency (EUE) is significantly positively correlated with net energy return, energy productivity, energy intensity, energy output, helio-thermal use efficiency, heat use efficiency and significant negatively correlated with specific energy and helio-thermal units. In result section, following statement is mentined=> The energy use efficiency and energy productivity were significantly higher with planting during 1st week of october along with application of 100 % RDF (14.24 and 0.45 kg MJ-1 respectively) (line no.576-580) .

6. “According to many researchers the inputs such as fuel, electricity, machinery, seed, fertilizer and chemical take significant share of the energy supplies to the production system in modern agriculture.” Please provide a reference for those studies.

Answer: References are inserted=> [11,12].(line no.

7. “Among the fertility levels, significantly higher gross return and net return was recorded with application of 200 % RDF and was on par with application of 150 % RDF due to higher grain and stover yield in these two fertility levels, whereas significantly lower gross return and net return was recorded with application of 100 % RDF due to its lower grain and stover yield. There is no significant difference with respect to B-C ratio” => The authors should discuss the reason(s) for such a result.

Answer: As both gross return and cost of production are increased at similar scale/ratio in all the treatment combinations and hence there is no significant difference with respect to B-C ratio. B-C ratio is not alone good indicator of best economics. Top priority should be allotted to net return and not to the B-C ratio. 

Discussion

1. The authors should provide caveats other researchers or readers should note in the interpretation of the results and overall implementation of the study.

Answer: In discussion section relevant referencs are being quated in support of study

2. The future direction for research should also be outlined and discussed.

Answer:Future direction for research is written as=> Validation and optimization of different sowing windows and fertility levels in different agro-climatic conditions over the years is required in order to address the dynamisms of environment and soil fertility.

3. The implications of the study particularly for farmers in South India should also be highlighted.

Answer: Unlike during rainy season, maize can be grown only under irrigation during winter season in entire South India. Variability of winter climatic condition at given year is minimal across maize area in south India. I feel these results may stand base for studying variability of sowing time and fertility across south India. However, results showed that early sowing during winter with (50 percent) more RDF could sustain higher productivity and economics of winter maize. In the title South India need to be deleted as my results may not address all agro-climatic zones and however they can be base for identifying location specific results.

Other general comments

1. Please include page numbers. This is one of the guidelines of the journal: "Include page numbers and line numbers in the manuscript file. Use continuous line numbers (do not restart the numbering on each page)."

2. The headings should also be numbered to allow for easy differentiation of main headings from subheadings.

3. Please, thoroughly check the whole manuscript for double spaces and full-stops.

4. Finally, subject the manuscript to a professional English proofreader and editor.

Answer: Page numbers and line numbers in the manuscript file are included. 

2. The headings and subheading are also numbered 

3. Thoroughly checked the whole manuscript for double spaces and full-stops.

4. Subject the manuscript to a professional English proofreader.

Reviewer #2: 

The manuscript is well articulated about optimization of sowing and nutrient management in winter maize. There are some typo and punctuation error in MS which needs to corrected. This needs some important revisions for more clarity:

Answer: All typo and punctuation error in text corrected.

In abstract: 

There is need to discuss interaction effects, if any. a line of recommendation needs to be added.

Answer: A line of recommendation added is=> From the overall interaction, it is recommended to plant winter maize during first fortnight of October with application of 150 percent RDF for sustaining higher maize productivity, energy output and economics in maize growing area of south India.(line no.43-46)

In line 1-5 of introduction there is two full stop (..) at two places, remove one.

Answer: Attended the suggestion.

In , Andra Pradesh (9.5 %), check spelling of Andhra Pradesh

Answer: Corrected as Andhra Pradesh

Exhaustive review needs to be included for identification of the problem for research. Include a para hypothesis clearly at the end of introduction.

Answer: Few more review has been added in support of problem identified. At the end of introduction, proposed hypothesis is clearly mentioned as => the testing hypothesis was initiated with objective of study the effect of sowing windows and fertility levels on growth and productivity of winter maize. And to work out the energy flow and production economics of winter maize cultivation under different sowing windows and fertility levels.(line no.117-127 and 896-905)

In Materials and Methods, 'With a spray of proclaim @ of 0.5 g litre-1', mention here technical name of it and also mention time of spray and how many spray were undertaken.

Answer: Suggastion is attended as=> With a spray of emamectin benzoate at 0.3 g litre-1 of water twice at 20 and 40 days after sowing the crop was protected against fall army worm and stem borer.Spray solution of 500l ha-1 was used at each time. (line no.163-165)

In Physiological efficiency (PE) formula, mention what A and F indicates.

Answer: Where, F– Fertilized plot; A – Unfertilized control plot (line no.214)

The bio-economics is new term used. How it is different from economics which is mostly reported as per studied parameter in the study. This needs to be economics in whole study.

Answer: bio-economics replaced with economics

In ' Planting during 1st week of October recorded significantly

higher GDD, PTU and HTU accumulation (1530.1 0C day, 17371.3 0C day hr and 11758.5

0C day hr respectively) and further it took significantly more number of days for

physiological maturity (120.0 days).' use insert symbol for mentioning the degree not superscript. Follow it in whole MS.

Answer: suggestion is implemented 

In 'Delayed planting caused shortening of growing degree days (GDDs)

accumulation during planting to physiological maturity[30]. ' The GDD is fixed, how this can be shortened, number of days can be lesser in delayed planting to increase in temperature may lead to forced maturity. This need to corrected accordingly.

Answer: The literature on heat indices showed that they all vary for attaining physiological maturity of crop depending on season and location. In similar way GDD vary to limited range to attain every phenological phases and maturity. In early sown maize took more days for maturity leads more accumulation of GDD and inturn result in more dry matter production and grain yield. In delayed sowing there was no forced maturity, however low temperature at germination delayed germination. several references [41,42,43] which proved that optimum temperature for maize germination, vegetative growth and flowering are 21 oC, 32 oC and 25-30 oC respectively. In present study late sown crop experienced lower temperature (18 oC, 23 oC and 23.1 oC during germination, growth and tasseling respectively) which affected crop yield. Further lesser availability of solar radiation (PTU) as a result of shorter day lengths in late planting (November) condition leads shorter growing period (Table 2) which reduced the vegetative growth, dry matter accumulation(Fig.1a) and finally the yield[42,43]. Forced maturity or reduction in yield of maize start when temperature surpasses 30 oC (Tesfaye et al., 2017).

(line no.435-466)

How the agronomic or physiological efficiency of the applied NPK were calculated without any control treatment in the study. This portion needs to be removed and partial factor productivity if studied may be included here.

Answer: Control treatment was existing in original data and is being inserted in the revised tables. 

The Fig. 1 is not a good representation of data. Needs to make as a bar diagram.

The Fig. 3 is also not giving any new information and needs to be dropped as it is not good representation of the data as well.

Answer: suggestions are being attended and original Fig. 1 deleted and made

---

## [Editor Report · Decision Letter 1]

2 Mar 2022

PONE-D-21-37373R1Productivity, nutrient use efficiency, energetics and economics of winter maize in south IndiaPLOS ONE

Dear Dr. Salakinkop,

Thank you for submitting your manuscript to PLOS ONE. After careful consideration, we feel that it has merit but does not fully meet PLOS ONE’s publication criteria as it currently stands. Therefore, we invite you to submit a revised version of the manuscript that addresses the points raised during the review process.

We look forward to receiving your revised manuscript.

Kind regards,

Paulo H. Pagliari

Academic Editor

PLOS ONE

Journal Requirements:

Additional Editor Comments (if provided):

please make sure all data is reported using appropriate units. Your grain yields are not correct in tables and figures. Review other data as well.
---

## [Author Response · Author response to Decision Letter 1]

29 Mar 2022

Response to Academic Editor and Reviewers' Comments dtd: 2nd March-22

Sub: Submission of revised covering letter and manuscript entitled “Productivity, nutrient use efficiency, energetics and economics of winter maize in south India” (PONE-D-21-37373)-reg 

Your email Ref: PONE-D-21-37373R1 dtd: 2nd March-2022

As advised by you on 2nd march-22, I am herewith furnishing following information.

1. Comments related to references: Please review your reference list to ensure that it is complete and correct. If you have cited papers that have been retracted, please include the rationale for doing so in the manuscript text, or remove these references and replace them with relevant current references. Any changes to the reference list should be mentioned in the rebuttal letter that accompanies your revised manuscript. 

Answer to comments : With related to references following points are brought your notice. Are also mentioned in rebuttal letter addressed to reviewers

1. Following paper was removed from both text and reference as they it was found not relevant 

Amanullah MM. et al. Influence of fertilizer levels and growth regulating substances on growth, nutrient use efficiency and yield of hybrid maize. 2010; Madaras Agric. J., 97(1-3): 68-72.

2. Following papers locations in the reference were changed in revised manuscript compared to original 

Devi S, Hooda VS and Singh J. Energy input-output analysis for production of wheat under different planting techniques and herbicide treatments. 2018; Int. J. Curr. Microbiol. App. Sci., 7: 749-760. (Line no. 754-756).

Vishwanatha VE. Integrated nutrient management in popcorn (Zea mays var. Everta). 2019; M. Sc. (Agri.) Thesis. Univ. Agric. Sci., Dharwad, India. (Line no. 677-678).

Vural Hasan, Efecan Ibrahim. An analysis of energy use and input costs for maize production in Turkey. J Food Agriculture Environ. 2012; 10 (2): 613-616. (Line no. 679-680).

Khokhar AK, Bawa SS, Singh S, Sharma V, Sharma SC, KumarV, et al. Tillage and nutrient-management practices for improving productivity and soil physicochemical properties in maize (Zea mays)–wheat (Triticum aestivum) cropping system under rainfed conditions in Kandi region of Punjab. 2018; Indian J Agron. 63(3): 278-284. (Line no. 681-684).

Singh SK, Singh RN, Ram US, Singh MK. Growth, yield attributes, yield, and economics of winter popcorn (Zea mays everta Sturt.) as influenced by planting time, fertility level, and plant population under late sown condition. J App Natural Sci. 2016, 8(3): 1438-1443. (Line no. 685-688). 

3. Following reference was not there in reference but quoted in text in original text. Therefore it was included in reference of revised manuscript.

Bevinakatti HP. Response of maize (Zea mays L.) to potassium silicate and silicic acid at different dates of sowing. M. Sc. (Agri.) Thesis. 2019, Univ Agric Sci. Dharwad, India. 

(Line no. 689-691).

4. Following six papers were added to original paper (both in text and reference) to complete the revised manuscript as per the suggestions of reviewers. 

Siddarta Hulamani, Salakinkop SR, Impact of sowing windows and fertility levels on growth and yield of winter maize in Northern Transitional Zone of Karnataka. J Farm Sci. 2021; 34(2): 142-148 (Line no. 841-843).

Suraj M, Somangouda G, Salakinkop SR. Yield and yield attributes of sweet corn (Zea mays L. saccharate) as influenced by the split application of nitrogen and potassium during kharif under protective irrigation. Journal of Entomology and Zoology Studies, 2020; 8(4): 361-364. 

(Line no. 892-895).

Thimme Gowda P, Halikatti SI, Manjunath SB. Thermal Requirement of Maize (Zea mays L.) as influenced by planting dates and cropping systems. Research Journal of Agricultural Sciences, 2013; 4(2): 207-210. (Line no. 896-908).

Girijesh GK., Kumara swamy AS, Sridhara S. Dinesh Kumar M, Vageesh TS, Nataraju SP Heat use efficiency and helio-thermal units for maize genotypes as influenced by dates of sowing under the southern transitional zone of Karnataka state. International J Science and Nature, 2011, 2(3): 529 – 533. (Line no. 899-902).

Rajput RP, Deshmukh MR,Paradkar VK. Accumulated heat units and phenology relationship in wheat as influenced by planting dates under late sown conditions. J Agron Crop Sci. 1987; 159:345-348. (Line no. 903-904).

Piper CS. Soil and Plant Analysis, 2002, Hans Publishers, Bombay, India.

2. Comments: please make sure all data is reported using appropriate units. Your grain yields are not correct in tables and figures. Review other data as well.

Answer: Grain and stover yields data in entire manuscript (tables,text and figures) are in same unit (kg/ha). Similarly other parameters are reviewed and are in same unit throughout the manuscript.

Following were the earlier answers to your reviewers’ advise (Ref: PONE-D-21-37373R1 

dtd: Thursday, Feb 17, 6:17 AM. (Just for your notice)

Abstract

1. “Maize area is rapidly spreading in south India in response…” Is maize spreading? With the way this was presented, it appears that the increase in the area where maize is cultivated is a natural phenomenon rather than the fact that farmers themselves are the ones cultivating and increasing the farm area dedicated to maize production.

Answer: Due to absence of major environmental impediments in winter, the desired field and crop operations can be planned and executed at the most desired time. And hence farmers are using this potential option to increase both area and production. Hence there is immense potential to increase the area under winter maize cultivation (line no 13-14).

2. “RDF(recommended dose of fertilizer)”. Normally, abbreviations should come after the full meaning is presented. Please reverse the order and include a space between RDF and the bracket.

Answer: Suggestion is incorporated in following sentence=> The present investigation encompassing different sowing windows with different fertility levels revealed that significantly higher winter maize productivity was achieved from either first and second week of October planting along with application of 200 % recommended dose of fertilizer (RDF) followed by 150 % RDF (line no 24).

3. The research gap, objective, methods used and contribution(s) of the study are missing or not clear in the abstract. Please rework your abstract to depict these important aspects of your manuscript.

4. Answer: Abstract has been revised by including research gap, objectives, methods used and contribution. The important information being added is as follow=> There were no planned field experiments to explore and optimize right time of sowing and quantity of fertilizer to be added previously due to presence of negligible winter maize area. Farmers’ used to cultivate maize as per their choice of sowing time with application of quantity of fertilizer recommended for rainy season maize. There was no efforts made towards working of economic analysis including energy budgeting. And hence the investigation was conducted with objective to explore the optimal planting period and fertilizer levels for winter maize through economic and energy budgeting. Planting windows (1st week of October, 2nd week of October, 3rd week of October, 4th week of October, and 5th week of October) and fertility levels (100 percent recommended dose of fertilizer (RDF), 150 percent RDF, and 200 percent RDF) were used as factors in Factorial Randomized Complete Block Design (RCBD) with three replications (line no 12-26 and 43-46).

.

5. 4. Please re-read and re-edit the whole abstract. Some sentences are incomplete while others are hard to understand. For example, “Also it recorded higher net returns and gross returns Whereas, energy use efficiency and energy productivity were higher with planting during first week of October along with application of 100 % RD”

Answer: Entire abstract has been re-edited taking into consideration of all suggestions (line no 12-26 and 43-46).

Introduction

1. Please provide a reference for this statement of fact: “The world's maize area is 192.50 million hectares, and it ranks first in production with 1,112.40 million metric tonnes.”

6. Answer: Reference is already quoted both in text and reference as =>[1] (line no 52).

2. Replace “it” with maize so that the reader can understand the sentence well: “After rice and wheat, it is India's third most popular crop”

Answer: Replaced as => After rice and wheat, maize is India's third most popular crop(line no 53).

3. “..Because of its photo-thermo-insensitive” => Correct the double full stops. Also, check for similar errors throughout the manuscript.

Answer: Double full stops replaced with single and similar mistakes were checked in entire text.

4. The authors tend to use maize and corn interchangeably in the manuscript. Please note that corn is used in North American English, while maize is used in British English. Moreover, the two terms can also have substantially different applications. I will suggest that you stick to one term, preferably maize since that is used more in the manuscript.

Answer: only one term “Maize” is used in entire text. 

5. “There are three distinct seasons for the cultivation of maize in India viz., Kharif, rabi in peninsular India and Bihar, and spring in northern India. Maize is predominantly a kharif season crop but in past few years winter maize has gained a significant place in total maize production in India[3].” Please rework this sentence because different font types and sizes were used. What do you mean kharif season? It may be good if you provide a short footnote describing that.

Answer: different font types and sizes were replaced with same font and size matching to remaining text. Kharif season is replaced with rainy season and rabi season is replace with winter season in entire text.

6. A whole of the first part of the introduction was dedicated to (winter) maize description, without arguments linking your keywords such as fertility, bioeconomics, energetic, and planting, etc. The authors should focus more on presenting arguments linking broader issues on productivity, nutrient use, energetics or bioeconomy during the winter season.

Answer: Following points were added to first part of existing introduction=>.

Nutrients use efficiency (NUE) shows the ability of crops to take up and utilize nutrients for higher productivity [4,58]. NUE depends on the plant’s ability to take up nutrients efficiently from the soil, but also depends on internal transport, storage and remobilization of nutrients. NUE of applied fertilizers may very low due to many reasons like surface runoff, leaching, volatilization, denitrification and fixation in the soil. The increased productivity show the higher nutrient use efficiency. The better planting date will provide the congenial environment to plants to uptake more nutrients so that productivity of crops is increased. In agriculture development, the energy audit of various resources plays a key role in resource management. Under the changing global climatic conditions and increasingly growing energy demands necessitate the development of a production system which utilizes less energy and produces more energy as output [4]. Fertilizer as input had the highest rate of energy equivalency of all the inputs used in maize production at 51.5 per cent[12] . The findings of Khokhar et al.[7] recorded higher input energy, output energy and energy balance in higher fertility levels and higher energy use efficiency and energy productivity in lower fertility levels (line no 72-87, 99-101). 

 In both rainy and winter season, higher gross return and net returns were obtained from maize sown early in the season compared to late sown crop [8- 9] (line no 84-87).

7. Just like the abstract, the research objective of the study is not clearly stated. Several questions should be answered in the introduction: what is the research or knowledge gap (s) that the study tries to fill, how was the gap(s) resolved and what contribution is the study making to the existing body of literature?

Answer: research objective, knowledge gap (s) and contribution is the study making to the existing body of literature was added at the end to introduction (line no 121-127). 

8. “There are a several of other factors that influence winter maize production and productivity; however, fertiliser management is one of the most important factors influencing maize growth and yields. Early maize planting can improve grain yields significantly, but other practises such as fertility can also trigger the yield[12]” => Please provide a reference after “… influencing maize growth and yields” These sentences are not clear. Is it fertilizer management that influences maize growth and yields or just fertilizer? In what direction would the factor influence maize growth and yields?

Answer: Reference is provided as directed > There are a several of other factors that influence winter maize production and productivity; however, fertiliser management is one of the most important factors influencing maize growth and yields [12]. Fertilizer management or recommendations on time and method of application of fertilizers have been already existed and it is similar to rainy season maize. However rate of application need to be worked out in order to meet higher winter maize productivity(line no 99-101).

9. “One of them is fertiliser control with care. Since, on the one hand, it accounts for more than half of the total input energy used in the maize production system in many cases, and on the other hand, it is the most important factor for proper plant growth and development.” More clarification is needed. How? Concerning the total input energy, are the authors referring to on-farm or off-farm activities, or both? Regardless, a reference is needed here.

Answer: First sentence clearly indicate that crop demand- driven approach is followed so that there is no huger for nutrients for showing its full yield potential. And at the same time applied fertilizer energy usage and efficiency were also worked out in order to economize the winter maize productivity. Both on-farm and off-farm activities were considered for working out total input energy referecnces cited are 6,15-16 (line no 108-118).

10. This sentence is incomplete: “As a result, excessive deployment, wastes resources and money while also exacerbating environmental problems[16].”

Answer: Same sentence is re-written> Excessive application of fertilizer results in wastes of resources and money while also exacerbating environmental problems [16] (line no 117-118).

11. Please present how the remaining part of the manuscript is structured at the end of the introduction.

Answer: After inclusion of all suggestion, the introduction flow is as follow.

a. Area, production and distribution of maize in world and India.

b. Importance of maize as feed,food and industry in-put

c. Winter maize productivity and area spread

d. Scope for increasing productivity of winter maize

e. Energy flow as input and output

f. Importance and quantity of fertilizer input requirement

g. Importance of sowing windows in winter

Materials and Methods

1. If you compare this “was conducted to investigate the response of winter maize to planting windows and fertility levels at the University of Agricultural Sciences, Dharwad (Karnataka)…” with “Hence experiment was conducted to explore the most congenial sowing period in Southern India” in the introduction, the purported objectives of the study are not the same. What is the broad objective of the study, and what specific objectives or questions does the manuscript answer? More specifically, south India was the focus in the title and the objective in the introduction, what is the reason(s) for using samples from the University of Agricultural Sciences? Would you consider your result as being valid and generalizable to other parts of Southern India?

Answer: Rainfall and temperature over the locations in south India vary during rainy season. However during winter, there is a need to raise the crop only by providing irrigation in entire maize growing areas’ of southern India. Further winter maize area in N-E India is also increasing and there productivity, crop duration and climate are different from that prevail in maize growing area of south India. Further Indian Institute of Maize Research (IIMR), apex body on maize research grouped entire maize growing area of south India in one zone (Zone-4- peninsular India). And hence to differentiate the regions, the south India is included in the title. General objective was to explore best sowing time and fertility levels. Specific objective were to study energitics and economics including winter maize productivity. These general and specific objectives been achieved for one location by taking into consideration of data of experiment from one location. I feel these results may stand base for studying variability of optimum sowing time and fertility across south India.

2. “The soil is classified as clay by the USDA soil textural classification table.” => Please provide a reference here. Aside from the soil type, more information about the soil composition should be provided.

Answer: The soil is classified as clay by the USDA soil textural classification table [17]. Soil composed of course sand, fine sand,silt and clay by 6.23,12.66,28.17 and 52.93 respectively(line no 135-137).

3. “Planting windows (1st week of October, 2nd week of October, 3rd week of October, 4th week of October, and 5th week of October) and fertility levels (100 percent recommended dose of fertiliser (RDF), 150 percent RDF, and 200 percent RDF) were used as factors in the experiment.” => Please present the actual dates and the year of the planting windows, instead of just weeks. This will enhance the reliability of the materials presented, as well as the replicability of your study.

Answer: First to fifth dates of sowing were 05-10-2019, 12-10-2019, 18-10-2019, 28-10-2019 and 06-11-2019 respectively. It is included the text (line no 142-144).

4. How many controls were considered in the study?

Answer: Interaction of November sowing with application of 100 per cent RDF is considered as control (line no 144-146)

5. Please first provide the full meanings before using acronyms e.g. for FYM and NUE.

Answer: Full meanings before acronyms are provided in entire text.

6. “The nutrients viz., nitrogen, phosphorus and potassium were applied @ 150 kg N ha-1, 65 kg P2O5 ha-1 and 65 kg K2O ha-1 for fertility level of 100 % RDF. Similarly for 150 % RDF, 225 kg N , 97.5 kg P2O5 and 97.5 kg K2O ha-1 was applied. And for 200 % RDF, 300 kg N, 130 kg P2O5 and 130 kg K2O ha-1 was applied through urea, diammonium phosphate (DAP) and muriate of potash (MOP) respectively. FeSO4 and ZnSO4 were applied @ of 25 kg ha-1.” => The authors should rework this aspect. It is not fully clear. What does viz stands for here?

Answer: Prefix The nutrients viz., is deleted and sentence has started with= > Nitrogen(line no 155)

After including your suggestion the sentence are redrafted as > All other treatment plots, including control plots, had well decomposed farm yard manure (FYM) @ 10 t ha-1 incorporated into soil two weeks prior to planting. The nitrogen, phosphorus and potassium were applied @ 150 kg N ha-1, 65 kg P2O5 ha-1 and 65 kg K2O ha-1 for fertility level of 100 % RDF. Similarly for 150 % RDF, 225 kg N, 97.5 kg P2O5 and 97.5 kg K2O ha-1 was applied. And for 200 % RDF, 300 kg N, 130 kg P2O5 and 130 kg K2O ha-1 was applied. Urea, diammonium phosphate (DAP) and muriate of potash (MOP) were used as a source of nitrogen, phosphorus and potassium respectively. Zinc and iron were applied in the form of FeSO4 and ZnSO4 at 25 kg ha-1. 

7. The authors tend to use @ instead of “at” throughout the manuscript. Please correct this. For example, “FeSO4 and ZnSO4 were applied @ of 25 kg ha-1.”

Answer: In entire text ‘@’ is replaced with ‘at’ in entire text.

8. Growing degree day: Please provide more description and justification on the reason(s) this method and the subsequent ones (e.g. Photothermal units, and Heliothermal units, etc.) are important. I assume that there are plausibly other methods that can be employed to attain similar results. Also, the processes involved in the implementation of these methods and their parameters are not clear. For GDD, which dates were the maximum and minimum temperatures reported, and how were these data captured. Providing such details would be good for non-expert readers and would also be crucial in aiding the readers to understand the framework followed in the study.

Answer: Many scientist used these GDD, PTU and HTU as a indicator of influence temperature and radiation on crop performance. There are several references which justify present results and I quate here some examples=> Swetha (2017) reported that early sowing of maize recorded higher grain yield compared to other delayed sowing due to higher accumulation of GDD, PTU and HTU. Sutton and Stucker (1974) reported that delayed sowing causes shortening of growing degree days (GDDs) accumulation during planting to physiological maturity. Similarly higher GDD accumulation was reported in early planted maize (Hugar, 2015). GDD, PTU and HTU units can be used to assess the suitability of a region for production of a particular crop, estimate the growth-stages of crops, predict maturity,best timing of fertilizer or pesticide application; estimate the heat stress on crops; plan spacing of planting dates to produce separate harvest dates (line no 173-178).

9. Also, are the formulas presented standardized? If yes, please provide references for them.

Answer: Formulas presented are referred universally and references for each formulae is already present (reference are[18-20]

10. Physiological efficiency (PE) indicates grain yield increase in kg per kg nutrient uptake from fertilizer[18]. And expressed in kilogram per hectare (kg ha-1). => Please rework this section, it is not clear.

Answer: Yes, rightly pointed out. And it is corrected as=>PE expression is grain yield in kg produced per kg of nutrient taken up from soil and expressed in kg kg-1 (line no 212)

11. Bio-economics: Is this the right term for what you intend to describe? Please note that bioeconomy is characterized by the utilization of biological resources from land and sea (e.g. plant and animal materials) for the production of a wide range of sustainable products and services aimed at driving economic growth through knowledge-based inventions and innovative biotechnological processes (line no 216 and 1).

Answer: Since we are dealing with biological entity (crop) and its products, the term bio-economics was used. And as per your suggestion it has been replaced with ‘economics’ in entire section.

12. “The price in USD of the inputs prevailed at the time.” It is not clear which inputs you are referring to here. Is the cost of the labor hours for preparing the land, and planting and harvesting also considered? The authors should which input costs were considered to avoid confusion for the readers.

Answer: All the inputs=> Land preparation, intercultivation, all applied fertilizers, FYM, seed, plant protection chemicals, irrigation, men and women wages right from sowing to harvesting , drying, processing and marketing of produce were included for working out cost of inputs(line no 118-121) 

13. “The energy balance was calculated using the data on input energy, output energy.” Please provide further information, this statement is not clear.

Answer: The energy balance in terms of net energy, energy use efficiecncy, energy productivity and specific energy was calculated using the data on input energy, output energy using the formula(line no 233-135)

14. The equations under Energetics are not well-formatted. Please rework them appropriately

Answer: Formatting will be done at copy editing stage since in MS wards, they tend to move. 

15. “The highest values were denoted by the letter ‘a,' which was followed by the alphabets for lower values (b, c, d, etc.).” => Be precise with the information provided in this sentence. The values of what?

Answer: Sentence is re-written as=> The highest mean values of all the crop and nutrient parameters statistically analyzed were denoted by the letter ‘a,' which was followed by the next alphabets for lower values (b, c, d etc.). At the 0.05 level of significance, mean values denoted by the same small letter in the column do not vary significantly (line no 254-258).

16. The authors should link each method to specific objectives to be answered. Mere presenting the methods is not enough, the rationale for choosing each method should be clear.

Linking of each method to specific objective has been done at appropriate places as =>. various thermal indices including growing degree days (GDD), Photothermal index (PTI) and heat use efficiency (HUE) for maize were calculated by using standard methods to know their influence on maize productivity. Similarly agronomic efficiency, physiological efficiency and recover efficiency were worked out to know relation between maize yield in response to nutrients applied. The energy balance in terms of net energy, energy use efficiecncy, energy productivity and specific energy were calculated taking into consideration of flow of inputs used for production in each treatment and out put obtained. Gross return, net return and B-C ratio were worked out to know the economics of maize production in each treatment ((line no 173-178, 207-209, 218-229).

 Results

1. Subheading: “Weather, GDD, PTU, HTU and days for physiological maturity.” Would a subheading “Descriptive statistics” be more appropriate as the heading here?

Answer: Subheading is changed as Descriptive statistics(line no 261) 

2. “During the winter cropping season (2019-2020), a total rainfall of 352.0 mm was received out of which 323.2 mm was received during the planting month (October) (Fig.4). The highest and the lowest maximum temperature were 31.8 ºC (February) and 28.5 ºC (December), respectively…” This information has been presented in the methodology section, so it feels more like repetition.

Answer: Yes and deleted from results section

3. Subheading: “Nutrient use efficiency in winter maize (NUE)” => NUE should be placed in the correct position immediately after efficiency. Rather, the author can do away with the abbreviation since it has been provided earlier in the manuscript

Answer: corrected as> Nutrient use efficiency (NUE) in winter maize (line no 312)

4. “Planting during 1st week of October along with application of 100 % RDF recorded significantly higher AEN, AEP and AEK (29.59, 68.29 and 68.29 kg kg-1 respectively) and it was found on par with planting during 2nd week of October along with application of 100 % and planting during 3rd week of October along with application of 100 % RDF.” => What does par mean in this context? Does it mean that there is no significant difference between the two planting periods when those parameters are compared? If so, please rework this throughout the manuscript as on par does not provide a statistical connotation.

Answer: It means that there is no significant difference between the two planting periods when those parameters (AEN, AEP and AEK) are compared. Your views are considered, however it is with respect to NUE and not with other attributes studied (line no.313-323)

5. “Interaction effect found non-significant with respect to PEP and PEK.” This sentence is not clear

Answer: Sentence is redrafted as> Whereas, interaction of sowing window and fertility level effect remained non-significant with respect to both PEP and PEK (line no 544-546).

6. Subheading Energetics: “Energetics of winter maize significantly influenced by planting windows and fertility levels.” It is not clear how this result was obtained. This points to my comment 15 on the methodology section. The authors should rework the authors' method section to depict how each objective is resolved and their results obtained.

Answer: Energetics and energy balance here are a general term and specifically they include variour components such as net energy, energy use efficiecncy, energy productivity and specific energy . And each was explained accordingly as per statistical analysis (line no553-632).

7. The subheadings in the methods and results sections are virtually the same. Authors should clearly distinguish the subheadings as they might become confusing for the readers to follow. For example, you have Bio-economics in the methods, results and discussion sections. In the results section, “Economic value of produced maize biomass” can be used as a subheading instead.

Answer: Suggestion is incorporated in result section

Discussion

1. “Late planting would lead to a lesser row number and less grain numbers in the rows of maize” How? Further clarification is needed.

Answer: It is just one of the references [38] to support low yield in delayed sowing. In my study average row number in each cob (15.1 to 17.4) remained significant among the treatments due to reduced cob girth. Sowing during 1st week of October along with application of 200 % RDF recorded significantly higher cob girth (17.6 cm) and it was on par with sowing during 2nd week of October along with application of either 200 % RDF or 150 % RDF (17.2 cm and 16.5 cm respectively).Whereas, sowing during 1st week of November along with application of 100 % RDF recorded lower cob girth (12.1 cm). Similarly, sowing during 1st week of October along with application of 200 % RDF recorded significantly longer cob (17.4 cm) and it was on par with all sowing windows except sowing during 1st week of November applied with all fertility levels. And hence there was less number seeds per seed row due reduced length of cob and seed size in terms of test weight. Data on these yield attributes will be presented, only after getting your suggestion. Above information is added (line no 416-427). 

2. “There was optimum climatic condition (maximum mean temperature 27.9 oC and minimum mean temperature 19 oC in October month) prevailed for crop sown during first and second week of October (early) planting”. This sentence is not clear.

Answer: It is clearly rewritten as => several references [41,42,43] which proved that optimum temperature for maize germination, vegetative growth and flowering are 21 oC, 32 oC and 25-30 oC respectively. In present study late sown crop experienced lower temperature (18 oC, 23 oC and 23.1 oC during germination, growth and tasseling respectively) which affected crop yield. Further lesser availability of solar radiation (PTU) as a result of shorter day lengths in late planting (November) condition leads shorter growing period (Table 2) which reduced the vegetative growth, dry matter accumulation (Fig. 1a) and finally the yield[42,43]. (Line no 442-448)

3. “The increased grain yield was due to improved yield attributes.” => This seems redundant and confusing. How is grain yield different from yield attributes? The authors should provide more explanations here.

Answer: Now, only one independent variable viz, grain yield has been mentioned instead of both grain yield and yield attributes. Generally all plant parameters including growth, yield parameters and yield are dependent variables for growth resources such as sowing time, fertility, variety, management and climatic etc variables (independent variables). And hence efforts have been made to explain magnitude of variation in independent variables (yield parameters and yield) as influenced by sowing windows and fertility levels (line no.463-466).

4. “Planting during 1st week of October along with application of 200 % RDF (W1F3) recorded significantly higher grain and stover yield” => The authors should remind the readers of the analytical methods employed to obtain the results being discussed. There seem to be some spacing issues here too. How are we sure that the higher grain and Stover yields are coming only from the RDF application? The descriptive analysis made is not sufficient to make such a conclusion.

Answer: At 5 per cent level of significance, planting during 1st week of October along with application of 200 % RDF (W1F3) recorded significantly higher grain and stover yield”. 

Principles of field experimentation such as randomization, replication and local control are followed while imposing treatments (on sowing window and fertility levels). Grain and stover yields are coming not only from the RDF application alone but also along with sowing window. However other growth resources are kept uniform (local control) for all treatment combinations ( line no 481).

5. “The energy use efficiency (EUE) was significantly positively correlated with net energy return, energy productivity, energy intensity, energy output, helio-thermal use efficiency, heat use efficiency and significant negatively correlated with specific energy and helio- thermal units” Which of the results indicate this in the results section?

Answer: It is just the references [74-75] to show universally that energy use efficiency (EUE) is significantly positively correlated with net energy return, energy productivity, energy intensity, energy output, helio-thermal use efficiency, heat use efficiency and significant negatively correlated with specific energy and helio-thermal units. In result section, following statement is mentined=> The energy use efficiency and energy productivity were significantly higher with planting during 1st week of october along with application of 100 % RDF (14.24 and 0.45 kg MJ-1 respectively) (line no.576-580) .

6. “According to many researchers the inputs such as fuel, electricity, machinery, seed, fertilizer and chemical take significant share of the energy supplies to the production system in modern agriculture.” Please provide a reference for those studies.

Answer: References are inserted=> [11,12].(line no.

7. “Among the fertility levels, significantly higher gross return and net return was recorded with application of 200 % RDF and was on par with application of 150 % RDF due to higher grain and stover yield in these two fertility levels, whereas significantly lower gross return and net return was recorded with application of 100 % RDF due to its lower grain and stover yield. There is no significant difference with respect to B-C ratio” => The authors should discuss the reason(s) for such a result.

Answer: As both gross return and cost of production are increased at similar scale/ratio in all the treatment combinations and hence there is no significant difference with respect to B-C ratio. B-C ratio is not alone good indicator of best economics. Top priority should be allotted to net return and not to the B-C ratio. 

Discussion

1. The authors should provide caveats other researchers or readers should note in the interpretation of the results and overall implementation of the study.

Answer: In discussion section relevant referencs are being quated in support of study

2. The future direction for research should also be outlined and discussed.

Answer:Future direction for research is written as=> Validation and optimization of different sowing windows and fertility levels in different agro-climatic conditions over the years is required in order to address the dynamisms of environment and soil fertility.

3. The implications of the study particularly for farmers in South India should also be highlighted.

Answer: Unlike during rainy season, maize can be grown only under irrigation during winter season in entire South India. Variability of winter climatic condition at given year is minimal across maize area in south India. I feel these results may stand base for studying variability of sowing time and fertility across south India. However, results showed that early sowing during winter with (50 percent) more RDF could sustain higher productivity and economics of winter maize. In the title South India need to be deleted as my results may not address all agro-climatic zones and however they can be base for identifying location specific results.

Other general comments

1. Please include page numbers. This is one of the guidelines of the journal: "Include page numbers and line numbers in the manuscript file. Use continuous line numbers (do not restart the numbering on each page)."

2. The headings should also be numbered to allow for easy differentiation of main headings from subheadings.

3. Please, thoroughly check the whole manuscript for double spaces and full-stops.

4. Finally, subject the manuscript to a professional English proofreader and editor.

Answer: Page numbers and line numbers in the manuscript file are included. 

2. The headings and subheading are also numbered 

3. Thoroughly checked the whole manuscript for double spaces and full-stops.

4. Subject the manuscript to a professional English proofreader.

Reviewer #2: 

The manuscript is well articulated about optimization of sowing and nutrient management in winter maize. There are some typo and punctuation error in MS which needs to corrected. This needs some important revisions for more clarity:

Answer: All typo and punctuation error in text corrected.

In abstract: 

There is need to discuss interaction effects, if any. a line of recommendation needs to be added.

Answer: A line of recommendation added is=> From the overall interaction, it is recommended to plant winter maize during first fortnight of October with application of 150 percent RDF for sustaining higher maize productivity, energy output and economics in maize growing area of south India.(line no.43-46)

In line 1-5 of introduction there is two full stop (..) at two places, remove one.

Answer: Attended the suggestion.

In , Andra Pradesh (9.5 %), check spelling of Andhra Pradesh

Answer: Corrected as Andhra Pradesh

Exhaustive review needs to be included for identification of the problem for research. Include a para hypothesis clearly at the end of introduction.

Answer: Few more review has been added in support of problem identified. At the end of introduction, proposed hypothesis is clearly mentioned as => the testing hypothesis was initiated with objective of study the effect of sowing windows and fertility levels on growth and productivity of winter maize. And to work out the energy flow and production economics of winter maize cultivation under different sowing windows and fertility levels.(line no.117-127 and 896-905)

In Materials and Methods, 'With a spray of proclaim @ of 0.5 g litre-1', mention here technical name of it and also mention time of spray and how many spray were undertaken.

Answer: Suggastion is attended as=> With a spray of emamectin benzoate at 0.3 g litre-1 of water twice at 20 and 40 days after sowing the crop was protected against fall army worm and stem borer.Spray solution of 500l ha-1 was used at each time. (line no.163-165)

In Physiological efficiency (PE) formula, mention what A and F indicates.

Answer: Where, F– Fertilized plot; A – Unfertilized control plot (line no.214)

The bio-economics is new term used. How it is different from economics which is mostly reported as per studied parameter in the study. This needs to be economics in whole study.

Answer: bio-economics replaced with economics

In ' Planting during 1st week of October recorded significantly

higher GDD, PTU and HTU accumulation (1530.1 0C day, 17371.3 0C day hr and 11758.5

0C day hr respectively) and further it took significantly more number of days for

physiological maturity (120.0 days).' use insert symbol for mentioning the degree not superscript. Follow it in whole MS.

Answer: suggestion is implemented 

In 'Delayed planting caused shortening of growing degree days (GDDs)

accumulation during planting to physiological maturity[30]. ' The GDD is fixed, how this can be shortened, number of days can be lesser in delayed planting to increase in temperature may lead to forced maturity. This need to corrected accordingly.

Answer: The literature on heat indices showed that they all vary for attaining physiological maturity of crop depending on season and location. In similar way GDD vary to limited range to attain every phenological phases and maturity. In early sown maize took more days for maturity leads more accumulation of GDD and inturn result in more dry matter production and grain yield. In delayed sowing there was no forced maturity, however low temperature at germination delayed germination. several references [41,42,43] which proved that optimum temperature for maize germination, vegetative growth and flowering are 21 oC, 32 oC and 25-30 oC respectively. In present study late sown crop experienced lower temperature (18 oC, 23 oC and 23.1 oC during germination, growth and tasseling respectively) which affected crop yield. Further lesser availability of solar radiation (PTU) as a result of shorter day lengths in late planting (November) condition leads shorter growing period (Table 2) which reduced the vegetative growth, dry matter accumulation(Fig.1a) and finally the yield[42,43]. Forced maturity or reduction in yield of maize start when temperature surpasses 30 oC (Tesfaye et al., 2017).

(line no.435-466)

How the agronomic or physiological efficiency of the applied NPK were calculated without any control treatment in the study. This portion needs to be removed and partial factor productivity if studied may be included here.

Answer: Control treatment was existing in original data and is being inserted in the revised tables. 

The Fig. 1 is not a good representation of data. Needs to make as a bar diagram.

The Fig. 3 is also not giving any new information and needs to be dropped as it is not good representation of the data as well.

Answer: suggestions are being attended and original Fig. 1 deleted and made as bar diagram

---

## [Editor Report · Decision Letter 2]

30 Mar 2022

Productivity, nutrient use efficiency, energetics and economics of winter maize in south India

PONE-D-21-37373R2

Dear Dr. Salakinkop,

We’re pleased to inform you that your manuscript has been judged scientifically suitable for publication and will be formally accepted for publication once it meets all outstanding technical requirements.

Kind regards,

Paulo H. Pagliari

Academic Editor

PLOS ONE
---

## [Editor Report · Acceptance letter]

1 Apr 2022

PONE-D-21-37373R2 

Productivity, nutrient use efficiency, energetic, and economics of winter maize in south India 

Dear Dr. Salakinkop:

I'm pleased to inform you that your manuscript has been deemed suitable for publication in PLOS ONE. Congratulations! Your manuscript is now with our production department. 

Kind regards, 

on behalf of

Dr. Paulo H. Pagliari 

Academic Editor

PLOS ONE